

# Future changes in surface ozone over the Mediterranean basin in the framework of the Chemistry-Aerosol Mediterranean Experiment (ChArMEx)

Nizar Jaidan[1], Laaziz El Amraoui[1], Jean-Luc Attié[1,2], Philippe Ricaud[1], and François Dulac[3]

[1]CNRM, Météo-France and CNRS, UMR 3589, Toulouse, France
[2]Laboratoire d'Aérologie, Université de Toulouse, UMR 5560, CNRS/INSU, Toulouse, France
[3]LSCE/IPSL, Laboratoire des Sciences du Climat et de l'Environnement, CEA-CNRS-UVSQ, Gif-sur-Yvette, France

*Correspondence to:* N. Jaidan (nizar.jaidan@meteo.fr)

**Abstract.**

In the framework of the Chemistry and Aerosol Mediterranean Experiment project (ChArMEx, http://charmex.lsce.ipsl.fr), we study the evolution of surface ozone ($O_3$) over the Mediterranean Basin (MB) with a focus on summertime over the time period 2000-2100, using the Atmospheric Chemistry and Climate Model Intercomparison Project (ACCMIP) outputs from

11 models. We consider three different periods (2000, 2030 and 2100) and the four Representative Concentration Pathways (RCP2.6, RCP4.5, RCP6.0 and RCP8.5) to study the changes in the future ozone trend and its budget. We use a statistical approach to compare and discuss the results of the models. We discuss the behavior of the models that simulate the surface $O_3$ over the MB. The ensemble mean of ACCMIP models simulates very well the annual cycle of surface $O_3$. Compared to measured summer surface $O_3$ datasets, we found that most of the models overestimate surface $O_3$ and underestimate its

variability over the most recent period (1990-2010) when independent observations are available. Compared to the reference period (2000), we found a net decrease in the ensemble mean surface $O_3$ over the MB in 2030 (2100) for 3 RCPs: -13% (-36%) for RCP2.6, -7% (-22%) for RCP4.5 and -11% (-33%) for RCP6.0. The surface $O_3$ decrease over the MB for these scenarios is much more pronounced than the relative changes of the tropospheric ozone burden. This is mainly due to the reduction in $O_3$ precursors and to the $NO_x$-limited regime over the MB. For the RCP8.5, the ensemble mean surface $O_3$ is almost constant

over the MB from 2000 to 2100. We show how the future climate change and the increase in $CH_4$ concentrations can offset the benefit of the reduction in emissions of $O_3$ precursors over the MB.



# 1 Introduction

Several modeling studies and assessments have evaluated the future evolution of chemical and dynamical processes and have shown that future changes in ozone precursors have a significant impact on the evolution of tropospheric ozone ($O_3$) and particularly surface $O_3$ (Butler et al., 2012; West et al., 2007). Among the changes is the stratospheric influx increase due, on one hand, to the global warming resulting from the accentuation of residual atmospheric circulation forced by climate change (Collins et al., 2003; Sudo et al., 2003; Butchart et al., 2006; Zeng et al., 2003) and, on the other hand, to the recovery of stratospheric ozone (Zeng et al., 2010; Kawase et al., 2011). The abundance of $O_3$ in the troposphere is controlled by various chemical and dynamical processes, sources such as chemical production, stratosphere-troposphere exchange (Danielsen, 1968), and sinks as chemical destruction and dry deposition. The magnitude of these processes depends on the abundance of $O_3$ precursors, the extent of climate change and also the geographical location. Tropospheric $O_3$ is an air pollutant, an efficient greenhouse gas and also the primary source of hydroxyl radicals that control the oxidation capacity of the troposphere. $O_3$ in the troposphere is produced by photochemical oxidation of methane ($CH_4$), carbon monoxide (CO) and volatile organic compounds (VOCs) in the presence of nitrogen oxides ($NO_x = NO + NO_2$). Moreover, the efficiency of photochemical reactions forming $O_3$ in the troposphere also depends on meteorological parameters such as temperature, radiation and precipitation (Monks et al., 2015; Jacob and Winner, 2009).

At the surface, $O_3$ is harmful to vegetation, materials and human health (Sandermann, 1996; Fuhrer and Booker, 2003; Lippmann, 1989; Brook et al., 2002) even at relatively low concentrations (Bell et al., 2004). High $O_3$ concentration is usually observed in summer period because meteorological conditions (high temperatures, weak winds, low precipitation) favor photochemical $O_3$ production (Meleux et al., 2007; Im et al., 2011).

The Mediterranean Basin (MB), surrounded by three continents with diverse pollution sources, is a region favoring the stagnation of pollutants and air pollution, in particular during summer (Schicker et al., 2010; Millán et al., 1996, 1997). This region is a hot-spot of climate change (Giorgi, 2006) that is due to its location and diversity of ecosystems. Gerasopoulos et al. (2005) showed that transport from the European continent was identified as the main mechanism that controls $O_3$ levels in the eastern MB. Akritidis et al. (2014) found significant negative $O_3$ trends between 1996 and 2006 over the MB due to the reduction of $O_3$ precursors emissions over continental Europe. A number of modeling studies have investigated the future change of surface $O_3$ in Europe including the MB (Wild et al., 2012; Langner et al., 2012; Colette et al., 2012; Fiore et al., 2009). The chemical regime over the MB and southern Europe presents a pronounced $NO_x$-limited regime (Beekmann and Vautard, 2010), except over maritime corridors and several major cities (e.g. Barcelona in Spain, Milano in Italy). In the $NO_x$-limited regime with relatively low $NO_x$ and high VOC, $O_3$ decreases with $NO_x$ anthropogenic emission reductions and changes little in response to VOC anthropogenic emission reductions, and the reverse occurs in the VOC-limited regime (Sillman, 1995). A number of studies dealing with the future changes in surface $O_3$ over the MB have been carried out at global and European scales. The assessment of the future changes in annual tropospheric $O_3$ at global scale has been done by Young et al. (2013) using a set of chemistry-climate models. At regional scale, Lacressonnière et al. (2014) studied the future changes in surface $O_3$ over Europe and the MB using a chemistry-transport model under the RCP8.5 scenario which corresponds to the





pathway with the highest greenhouse gases emissions, leading to a radiative forcing of the order of 8.5 W.m$^{-2}$ at the end of the 21st century. The limited number of models and scenarios used in different studies increases the uncertainty and weakens the reliability of the results. In this paper, we analyse simulations performed from a set of chemistry-climate models under the four Representative Concentration Pathways (RCP2.6, RCP4.5, RCP6.0 and RCP8.5; Van Vuuren et al., 2011), defined in

section 2.1, to investigate the future changes in surface O$_3$ over the MB, under a wide range of future projections. We highlight the impact of different factors contributing to surface O$_3$ change : emissions, meteorological and chemical parameters. This will also enable a better understanding of the effect of reducing O$_3$ precursors on the future evolution of surface O$_3$.

In the framework of the Chemistry and Aerosol Mediterranean Experiment project (ChArMEx, http://charmex.lsce.ipsl.fr), we focused on future changes in surface O$_3$ from 2000 to 2100 above the MB using model outputs from the Atmospheric

Chemistry and Climate Model Intercomparison Project (ACCMIP; Lamarque et al., 2013). ACCMIP consists of a series of time slice experiments aiming at studying the long-term changes in atmospheric composition between 1850 and 2100. ACCMIP was designed to feed the Intergovernmental Panel on Climate Change (IPCC) Fifth Assessment Report (AR5) and targets the analyses of the driving forces of climate change in the simulations being performed in the 5th Coupled Model Intercomparison Project (CMIP5) (Taylor et al., 2012). This paper is organised as follows. In Section 2, we provide a summary of the datasets

used in this study, as well as the analysis approach. Section 3 focuses on the evaluation of the present-day (1990-2010) surface O$_3$ simulations compared to independent observations. In Section 4, we explore the future change in surface O$_3$ for the periods 2030 and 2100 over the MB and discuss the various drivers affecting this change such as meteorological parameters and O$_3$ precursors. Conclusions are given in Section 5.

## 2   Datasets and analysis approach

In this section, we provide some details about the ACCMIP models, the scenarios and the observations used in this study, followed by a general description of the analysis approach.

### 2.1   ACCMIP models and observations

We used the data from 11 models from the ACCMIP Experiment. Note that model outputs are not available for all scenarios and periods (see Tables 1 and 2). Some ACCMIP models (namely CICERO-OsloCTM2, MOCAGE and MIROC-CHEM) are

excluded in this study due to the absence of sufficient and required model outputs.

A general evaluation and a detailed ACCMIP model description are provided in Lamarque et al. (2013). The models are driven by sea-surface temperature (SST) and sea-ice concentrations (SICs). The complexity of chemical schemes varies considerably between models, from the simplified schemes of CESM-CAM-Superfast (16 species) to the more complex schemes of GEOSCCM (120 species). The differences between models mostly come from the degree of representation of non-methane

hydrocarbon (NMHCs) emissions and chemistry in the models. The representation of stratospheric chemistry is included in the models, excepted in HadGEM2, LMDz-OR-INCAand STOC-HadAM3 and UM-CAM. LMDz-OR-INCA uses a constant (in time) stratospheric ozone climatology (Li and Shine, 1995), whereas the other models without detailed stratospheric chem-



istry uses the time varying stratospheric $O_3$ dataset of Cionni et al. (2011). All anthropogenic and biomass burning emissions are specified for all models, however the natural emissions are differently specified for the different models. In many cases, different models share several aspects such as dynamical cores, physical parameterizations, convection or the boundary layer scheme, but differ much in the number of chemical reactions. Consequently, all the models used in our study are considered as

distinct according to Lamarque et al. (2013).

A new set of future projections according to four scenarios named as Representative Concentration Pathways (RCPs) was released for CMIP5 (Moss et al., 2010). The RCPs are named according to radiative forcing (RF) target level for 2100. The radiative forcing estimates are based on the forcing of long-lived and short-lived greenhouse gases and other forcing agents. The RCPs are four independent pathways developed by four separate Integrated Assessment Modeling groups (IAMs). The

socio-economics assumptions underlying each RCP are not unique, the four selected RCPs were considered to be representative of a larger set of scenarios in the literature, and include one mitigation scenario leading to a very low forcing level (RCP2.6) which assumes a peak in RF at $3.0$ W.m$^{-2}$ in the early 21st century before declining to $2.6$ W.m$^{-2}$ in 2100 (Van Vuuren et al., 2006, 2007), two medium RF stabilization scenarios (RCP4.5; RCP6.0), which stabilize after 2100 at $4.5$ W.m$^{-2}$ and $6.0$ W.m$^{-2}$, respectively (Fujino et al., 2006; Smith and Wigley, 2006; Wise et al., 2009; Hijioka et al., 2008) and one very

high baseline emission scenarios (RCP8.5) which assumes an increasing RF even after 2100 (Riahi et al., 2007). In a first phase, ACCMIP historical simulations (Hist) were carried out covering the pre-industrial period to the present day (Lamarque et al., 2010). Secondly, ACCMIP simulations were performed based on a range of RCPs (Van Vuuren et al., 2011) to cover 21st century projections. Ozone precursor emissions from anthropogenic and biomass burning sources were taken from those compiled by Lamarque et al. (2010) for the Hist simulations, whereas emissions for different RCPs simulations are described

by Lamarque et al. (2013). The four RCPs include reductions and redistribution of $O_3$ precursor emissions in future projections except for $CH_4$. Natural emissions, such as CO and VOCs from vegetation and oceans, and $NO_x$ from soil and lightning, were determined by each model group. In this study, we use available surface $O_3$ observations based on the gridded observations given by Sofen et al. (2015) in order to evaluate uncertainty related to model simulations. Sofen et al. (2015) built a consistent gridded dataset for the evaluation of chemical transport and chemistry-climate models from all publicly available surface $O_3$

observations from online databases of the modern era: the World Meteorological Organization (WMO) Global Atmospheric Watch (GAW), Cooperative Programme for Monitoring and Evaluation of the Long-range Transmission of Air Pollutants in Europe (EMEP), European Environment Agency Air-Base (EEA), US Environmental Protection Agency Clean Air Status and Trends Network (US EPA CASTNET), US EPA Air Quality System (AQS) Environment Canada's Air and Precipitation Monitoring Network (CAPMoN), Canadian National Air Pollution Survey Program (NAPS) and Acid Deposition Monitoring

Network in East Asia (EANET). The surface $O_3$ data used at global scale are built from 2531 sites, mostly located (97%) in the northern mid-latitudes between 22°N and 69°N mainly in North America and Western Europe (Sofen et al., 2015). Data are averaged within a global grid of 2° x 2°. We use averages from hourly $O_3$ data on a monthly basis from 1990 to 2010 (see Fig. 1).



## 2.2    Analysis approach

In this study, we analyze the present day and future simulations performed by the ACCMIP models over the MB, in order to assess the surface $O_3$ evolution in a context of climate change. We use the four scenarios (RCP2.6, RCP4.5, RCP6.0 and RCP8.5) and focus on three periods: a reference period (REF) which corresponds to the 2000 time slice from the historical

scenario and two future periods in both the short and long term, corresponding to the 2030 and the 2100 time slices, respectively. The number of years simulated for each time slice mostly varied between 8 and 16 years for each model (see Fig. 1). The number of scenarios available is between 1 (GEOSCCM) and 4 (LMDz-OR-INCA, E2R-GISS, GFDL-AM3 and NCAR-CAM3.5) (see Table 2 showing the available scenarios as well as the meteorological and chemical parameters for each model). Note that this study is composed of two parts. The first part consists of a model assessment based on the REF period, in which

we compare the outputs of different models to a set of available surface $O_3$ observations based on the gridded observations given by Sofen et al. (2015). We use several statistical diagnostics to assess the performances of different model outputs. The individual model performances are calculated using coincident observations, while the ACCMIP ensemble mean is compared to the averaged observations over the period (1990-2010). For the evaluation of the different models, we use a complete set of metrics (see Table 3): the coefficient of variation ratio (CvR), the correlation coefficient (R), the normalized mean biases

(NMB), the Mean Bias (MnB), the Mean Absolute Gross Error (MAGE) and the Root Mean Square Error (RMSE). In addition to these metrics, we use two unbiased symmetric metrics introduced by Yu et al. (2006) that are found to be statistically robust and easier to interpret: the Normalized Mean Bias Factor (NMBF) and the normalized mean absolute error factor (NMAEF). The aim is to better understand the behavior of each model that simulate annual and summer surface $O_3$ in recent conditions. The second part is dedicated to the study of the future evolution of surface $O_3$ in summer with a link to meteorological

variables (temperature, humidity and precipitation) and $O_3$ precursors at the surface ($CH_4$ concentration, CO, VOCs and $NO_x$ emissions). The study is focused on June, July and August (JJA) except for the investigation of the annual cycle of surface ozone over the MB (section 3.1). We averaged the available output simulations in summertime (JJA), and over the box representing the MB domain included in the Mediterranean region (see Fig. 2). This future projection is compared to the REF period, using the box-whisker plots by specifying outliers with interquartile rule for outliers (IQR). In order to highlight the regions with a

significant change in surface ozone, as well as to evaluate the statistical significance of our results, we use the Student t test for a 95% confidence level. The future evolution of the $O_3$ budget is also discussed.

## 3    Evaluation of present-day surface ozone from ACCMIP models

ACCMIP model simulations have been extensively evaluated on a global scale by Lamarque et al. (2013) and Young et al. (2013). In this paper, we study the behavior of each model that simulates surface $O_3$ and we focus on the MB. We compare the

REF ACCMIP simulations (see Table 1) to surface $O_3$ observations based on the gridded observations given by Sofen et al. (2015). Our evaluation includes four parts: (1) evaluation of the annual cycle of surface $O_3$ over the MB; (2) discussion and evaluation of the modeled ACCMIP mean surface $O_3$ in summer; (3) comparison between simulated and observed summer



surface $O_3$ with a distinction between mainland and sea; and (4) evaluation of models with a wide range of metrics and comparison of their performances between the regional and the global scales.

## 3.1 Annual cycle of surface ozone over the Mediterranean Basin

Figure 3a compares the annual cycle of surface $O_3$ from the ACCMIP ensemble and the ACCMIP annual mean against gridded observations. Most models are in agreement with the observed annual cycle showing a maximum in summer and a minimum in winter, except CESM-CAM-superfast, which shows a decrease in ozone during summer to reach a concentration equal to the observed surface ozone, and shows strong overestimations in other seasons. We also observe a general overestimation of the modeled surface $O_3$ that is more pronounced in summer and particularly for GISS-E2-R, GEOSCCM and LMDz-Or-INCA with a mean bias of 9.20 to 20.25 ppbv compared to observations. The ACCMIP mean simulates appropriately the surface $O_3$ with a consistent positive bias varying between 6.44 and 14.62 ppbv. The CMAM model reproduces very well the annual cycle (Fig. 3a). Figure 3b shows the Taylor diagram (Taylor, 2001) which compares the annual cycle of surface $O_3$ of different ACCMIP models to coincident observations, averaged over a period from 8 to 16 years and between 1990 and 2010 (see Fig. 1). This diagram allows us to objectively compare the simulated and the observed annual cycle. In the Taylor diagram, the simulated patterns that agree the best with the observations should be close to the open circle marked "Obs" on the x-axis (see Fig. 3b). The correlation coefficient (R) between simulated and observed annual cycle of surface $O_3$ is generally greater than 0.75 for most of the models except for LMDZ-OR-INCA and CESM-CAM-superfast ($0.55 <$ R $< 0.75$). GISS-E2-R and GEOSCCM reach a correlation coefficient of 0.8. For the other models, the correlation coefficient exceeds 0.92. GEOSCCM, NCAR-CAM3.5 and GFDL-AM3 present a normalized standard deviation close to 1. The ACCMIP mean simulates very well the surface $O_3$ and shows better performance than most of the other models except GFDL-AM3 with a correlation coefficient of 0.93 and a normalized standard deviation of 0.87. In conclusion, most of the models are in agreement with the observations in terms of the annual cycle with a correlation coefficient greater than 0.8. The models that represent the best the annual cycle of surface $O_3$ are: GFDL-AM3, NCAR-CAM3.5 and CMAM.

## 3.2 Modeled ACCMIP summer mean surface ozone

Figure 4a shows the ACCMIP multi-model ensemble mean of the summer surface $O_3$ over the REF period. The general features, with higher $O_3$ concentrations over the MB and the Middle East region, are observed, exceeding an average of 60 ppbv in the center of the MB, than over continental Europe and Northern Africa ($\approx$40 ppbv). Several modeling studies have already shown this gradient in $O_3$ concentration between land and sea (Lelieveld and Dentener, 2000; Zeng et al., 2008; Langner et al., 2012; Lacressonnière et al., 2012; Safieddine et al., 2014). A minimum in surface ozone is simulated over the North-Western Europe region, which corresponds to a VOC-limited regime in summertime, unlike the MB which is characterized by a $NO_x$-limited regime as shown by Beekmann and Vautard (2010). This means that the ACCMIP ensemble mean respects the spatial variability of ozone related to the chemical regime. All models capture this variability in surface $O_3$ concentrations (not shown). Figure 4b shows the ACCMIP ensemble standard deviation (sd) of the summer surface $O_3$ over the period 1990-2010. The different models are generally in agreement over the MB except over the Ligurian Sea (southern Po Valley, Italy




and around Marseille, France) with $sd > 13\,ppbv$ (Fig.4b). This region is characterized by a high density of anthropogenic and natural emissions and by the frequent occurrence of stagnant meteorological conditions (Silibello et al., 1998; Finzi et al., 2000; Martilli et al., 2002; Dufour et al., 2005; Kalthoff et al., 2005; Meleux et al., 2007). Vautard et al. (2007) show that the overestimation of simulated ozone concentrations in the Po Valley is probably due to the excessive stagnation of winds, and that

the ability of models to simulate acute episodes is strongly variable in the Po Valley explaining the variability between models. Figure 5a shows the ACCMIP ensemble mean bias of the summer surface $O_3$ over the period 1990-2010. Colored circles indicate the representative gridded observations. The black circles represent mainland and large islands labeled as "land". The green circles represent cell box included in the sea domain and are mainly represented with small islands labeled as "sea". The ACCMIP ensemble mean overestimates surface $O_3$ over the sea and central Europe (Fig. 5a). However, the mean bias is

negative in Crete and in some regions in Spain. The ACCMIP ensemble mean of absolute error (Fig. 5b) shows an absolute error distribution similar to the distribution of the mean bias with a maximum absolute error of 12 ppbv over the central MB, central and Eastern Europe and an absolute error of 5 ppbv in Crete and Cyprus. Our study is consistent with various modeling studies that have shown that models overestimate surface $O_3$ observations at northern mid-latitudes (Young et al., 2013; Parrish et al., 2014).

### 15   3.3    Comparison of present-day modeled surface ozone in summer with coincident observations

Figure 6 shows the scatter plot of summer surface $O_3$ observations versus the different ACCMIP model simulations for a period ranging from 8 to 16 years in the time period between 1990 and 2010. Cyan color dots and red color dots correspond to observations over sea and over land, respectively as defined in section 3.2, one dot per grid per summer (JJA) of year. In this figure, the elliptical curve represents an estimation of the scatter plot in order to better compare the different classes. The

shape of the ellipse shows the distribution of colored dots in each class. For example, a flat ellipse following the y-axis means a small variability of the model compared to the observations. The bias and the difference of dispersion between models and observations are quantified by the normalized mean biases (NMB) and the coefficient of variation ratio (CvR), respectively (see Table 3). Compared to observations, we note that most models overestimate surface $O_3$ (dots) lying down above the diagonal reference line. The NMB varies from -5% to 44% for land surface and from 13% to 73% for sea surface. CMAM

and CESM-CAM-Superfast are the models with the lowest NMB over land (-5% and 0%) and over sea (18% and 13%), respectively. Conversely, GISS-E2-R is the model with the highest NMB: 73% over the sea and 44% over the land. We note that, compared to observations, most models tend to overestimate surface $O_3$ over sea, and are better represented over land in southern Europe than over the Mediterranean Sea. In the same way, several studies (Ganzeveld et al., 2009; Coleman et al., 2010) suggested that models are deficient in terms of dry deposition of gaseous species over oceans. This underestimation due

to a misrepresented turbulence within the models could lead to an $O_3$ overestimation. The general overestimation of surface $O_3$ by models compared to observations is likely due to a deficiency in VOCs emissions (Young et al., 2013). Moreover, we found that all the models used in our study present a smaller variability compared to the observations, but more pronounced over sea. The CvR that shows the dispersion ratio between observations and models varies from 1.4 to 3.3 for land surface and



from 3.6 to 5.5 for sea surface. GISS-E2-R, CMAM and GEOSCCM are the closest to the observations in terms of variability with a CvR less than 1.8 and 4.0 for land and sea, respectively.

## 3.4 Model performance metrics

A comparison of tropospheric $O_3$ between ACCMIP models and observations from ozonesondes and space-borne instrument is provided by Young et al. (2013). It shows that the ACCMIP ensemble performances to simulate tropospheric $O_3$ vary between different regions over the world. In our study, we use the ACCMIP simulations of surface $O_3$ over a specific region, namely over the MB. We compare the performances of the models at regional and global scales. Figure 7 shows the ACCMIP model performances terms of R, MnB, MAGE, RMSE, NMBF, and NMAEF, based on spatio-temporal comparison of surface $O_3$ between ACCMIP model simulations and coincident observations over the REF period. Rows and columns represent individual metrics and models, respectively. Each cell contains the value of a corresponding metric and a color indicating the performance of the model, from white (the closest to the observations) to dark red (the farthest from the observations). Each metric is calculated at regional and global scales. Comparing the two colored Tables (Fig. 7), we note that the color distribution is on average similar. This means that there is no significant difference in the model performances regarding the scale (global vs regional Mediterranean) except for the GEOSCCM model whose performances are better at global than at regional scale. The correlation coefficients (R) are similar and vary from 0.28 to 0.58 and from 0.36 to 0.60 for the global and regional scales, respectively. Note that EMAC, GEOSCCM and CESM-CAM-superfast have a higher bias and error at regional scale, particularly for GEOSCCM with a NMBF and a NMAEF of 0.45 and 0.47 against 0.3 and 0.35 at the global scale, respectively, unlike for the other models that have a slightly better score at the regional scale. GISS-E2-R is obviously the farthest model from the observations with a NMBF and a NMAEF greater than 0.52 and with a R of 0.39 over the MB. The closest model to observations is CMAM with a NMBF close to zero and a NMAEF less than 0.24. GFDL-AM3 presents a correlation coefficient of 0.66 over the MB, but the error and bias are greater than 14 ppbv. Parrish et al. (2014) show that outputs from three current ACCMIP models overestimate ozone mixing ratios at northern mid-latitudes, on average by 5 to 17 ppbv in the year 2000.

In conclusion, this evaluation shows that the models are different in terms of performances and most of the models overestimate the surface $O_3$. The bias is positive for all models except CMAM at the global scale. The model performances do not significantly change on average from the global to the regional scale (MB) over the REF period. Quantifying models uncertainty by comparison with observations in recent past will help us to estimate their accuracy in the future projections.

## 4 Future changes in summer $O_3$

In this section, we study the future changes in surface $O_3$ and its budget over the MB in 2030 and 2100 compared to 2000. We also discuss the factors that could impact future trends in surface $O_3$: meteorological variables (temperature, specific humidity and precipitation), ozone precursors at the surface ($CH_4$ concentration, CO, VOCs and $NO_x$ emissions), future climate change. We use all available data from the 11 ACCMIP models (see Table 2) among which 8 models have been evaluated in section 3. Our study focuses mainly on the ACCMIP ensemble mean, which is representative of the ACCMIP ensemble (found to



be close to observations). The future changes in surface $O_3$, ozone precursors and meteorological variables are averaged over the domain shown in Fig. 2. The entire study is focused on June, July and August (JJA) to be representative of the summer conditions.

## 4.1 Future changes in meteorological parameters

For each of the 4 RCPs, Fig. 8 shows the mean change in meteorological parameters from the ACCMIP models over the MB for the JJA period 2000, 2030 and 2100. The number of available models for each period is varying according to the different scenarios, but it is the same between 2030 and 2100 for each scenario except for RCP6.0 with one more model (GEOSCCM) in 2100 compared to the 2030 simulations (see Table 2). The general trend in temperature from 2000 to 2100 is increasing (Fig. 8a), and the amplitude depends on the scenario and the period. An increase in temperature of 0.4-1.1 K and 0.1-4.5 K is noted for the period 2000-2030 and 2030-2100, respectively. This increase depends linearly on the radiative forcing. Outlier models for temperature are GISS-E2-R and STOC-HadAM3 for RCP2.6 (2030), GISS-E2-R for RCP6.0 (2030) and CESM-CAM-Superfast which show a strong maximum in temperature in the RCP8.5. Inter-model variability grows as a function of the increase in RF and is generally greater for 2100 than for 2030. Temperature increases on average by about 0.8 K for RCP2.6 and by 5.5 K for RCP8.5, between 2000 and 2100. In general, an increase in temperature favors biogenic emissions (mainly isoprene, a biogenic precursor of ozone) and favors photochemical reactions (Derwent et al., 2003). In addition, the general trends in specific humidity (Fig. 8b) and temperature are similar. This can be interpreted as a result of evaporation, knowing that the MB will be affected by climate change and particularly exposed to high temperatures. The NCAR-CAM3.5 is an outlier in terms of specific humidity. It presents a decrease in the specific humidity between 2030 and 2100 unlike the other models. Inter-model variability is greater for RCP2.6 and RCP6.0 than for the other scenarios, which is probably due to the uncertainty in the temperature change for the RCP2.6 and perturbation due to GEOSCCM Model, that shows a minimum of humidity in 2100 for RCP6.0. Several studies have shown that humidity is the most important meteorological factor affecting OH and $CH_4$ lifetimes (Spivakovsky et al., 2000), which are involved in the chemical production of $O_3$.

In general, precipitations decrease for all RCPs (Fig. 8c) and the decrease is more pronounced for RCP6.0 and RCP8.5. Decreased precipitations mean decreased cloudiness and increased solar radiation, which drive the photochemical production of $O_3$ (Langner et al., 2005). Available REF simulations for the specific humidity and precipitation are represented only by 3 and 2 models, respectively (see Table 1). For future simulations, the specific humidity and precipitation are represented by at least 5 models among all the scenarios. As a conclusion, the ACCMIP mean surface temperature increases during the 21st century for the four RCPs, according to the RF. The surface specific humidity increases over the MB as a response to the rise in surface temperature and precipitation decreases for scenarios that have the highest RF (RCP6.0 and RCP8.5).

## 4.2 Future changes in ozone precursors

One of the strong asset of the ACCMIP experience is that ozone precursors have been specified for all models. However, the biogenic emissions were not specified. Their estimates depend on each modeling group, which can add much to the inter-model variability in addition to differences in model complexity and parameterizations. Figure 9 shows the mean change in ozone



precursors (surface $CH_4$ concentration, VOCs, CO and $NO_x$ emissions) in the ACCMIP models averaged over the MB over the JJA period of 2000, 2030 and 2100 time slices. $CH_4$ concentration at the surface decreases over the MB (Fig. 9a) between 2000 and 2030 by 10% for RCP2.6 and increases for RCP4.5, RCP6.0 and RCP8.5 by 6%, 6% and 27%, respectively. Conversely, between 2030 and 2100, the average concentration of $CH_4$ at the surface over the MB decreases by 21%, 12% and 6% for

RCP2.6, RCP4.5 and RCP6.0, respectively. However, in the same period for RCP8.5, surface $CH_4$ concentration increases by 73%. Inter-model variability of $CH_4$ is small relative to the total change for all RCPs. We also note that the total change in $CH_4$ concentration over the MB is almost the same between RCP4.5 and RCP6.0, despite a significant difference in RF mainly due to the difference in the concentration of $CO_2$ between these two scenarios. This detail is important in the interpretation of the difference in the surface $O_3$ concentration between RCP4.5 and RCP6.0, knowing that long-term change in $CH_4$ induces

changes in $O_3$ (West et al., 2007). The maximum and minimum $CH_4$ concentrations observed in the four scenarios come from GISS-E2-R and LMDz-OR-INCA, respectively and can be considered as outliers according the interquartile range rule (IQR). In fact, these two models are the only ones that do not prescribe $CH_4$ concentrations in RCPs simulations (Young et al., 2013).

Figure 9b presents the evolution of total VOCs emissions between 2000 and 2100. We note that the inter-model variability is high. This is mainly due to two factors: (1) The VOC module is different from one model to another. In other words, some

models have more VOC species than others, and especially isoprene is not included in a few models. (2) The second factor is that the biogenic emissions are not specified in the ACCMIP experiment. VOCs emissions are mainly from biogenic origin, which explains this difference (Lamarque et al., 2013; Young et al., 2013). Regarding the REF simulations, the multi-model average of VOCs is high, due to the limited number (only 2) of available models (Table 1). This value is not representative and we cannot compare it with values from future simulations of VOCs. Multi-model average of CO (Fig. 9c) decreases from 2000

to 2100 for all the RCPs, by 57%, 57%, 61% and 70% for RCP2.6, RCP4.5, RCP6.0 and RCP8.5, respectively. This reflects the pollutants reduction policy that was implemented for the four scenarios in the integrated assessment model (IAMs) (Van Vuuren et al., 2011). The inter-model variability is relatively high, likely due to the difference between models in the representation of natural emissions from vegetation and ocean as well as in the complexity of their chemical schemes. Outliers are HADGEM2 for RCP2.6, RCP4.5, RCP8.5 and GEOSCCM for RCP6.0, which correspond to a maximum of CO emission. Figure 9d shows

that $NO_x$ emissions generally decrease for the four RCPs. This decrease from 2000 to 2100 is more pronounced for RCP2.6 and RCP6.0 by 61% and 72%, respectively, than for RCP4.5 and RCP8.5 by 41% and 34%, respectively. In addition, the inter-model variability is relatively small. HADGEM2 is an outlier, representing the maximum of concentration in RCP2.6, RCP4.5 and RCP8.5. Other outliers are CESM-CAM-Superfast for RCP2.6 (2100) and RCP6.0 (2030), and EMAC for RCP4.5 (2100) and RCP8.5 (2100). NCAR-CAM3.5 and GFDL-AM3 represent the minimum for RCP2.6. We identified outliers models which

can adversely affect the quality of our results, but in terms of the future evolution, all models have similar trends.

In conclusion, the emissions of CO and $NO_x$ decrease linearly during the 21st century for the four RCPs, reflecting the emission reduction policy. The change in VOCs is not significant given the inter-model variability. The surface $CH_4$ concentration increases between 2000 and 2030 for RCP4.5, RCP6.0 and RCP8.5, and decreases for RCP2.6. However, the surface $CH_4$ concentration increases drastically for RCP8.5 between 2030 and 2100 and decreases for the other scenarios over the

same period.



### 4.3 Future changes in surface ozone

Figure 10 shows the mean change in summer surface $O_3$ between 2000 and 2100 over the MB. Compared to 2000, the relative changes for the summer surface ozone over the MB domain (see Fig. 2) in 2030 (2100) for the different RCPs are: -13% (-36%) for RCP2.6, -7% (-22%) for RCP4.5, -11% (-33%) for RCP6.0 and -0.4% (0.2%) for RCP8.5. The models with the most

5 pronounced decrease are GISS-E2-R, GFDL-AM3 and NCAR3.5. These models are bias high compared to the observations as seen in section 3.4 (Fig. 7). However, the models are generally in agreement in terms of $O_3$ future decrease between 2000 and 2100, except for the RCP8.5. Young et al. (2013) show that the relative changes for the global tropospheric ozone burden in 2030 (2100) are: -4%(-16 %) for RCP2.6, 2%(-7 %) for RCP4.5, 1%(-9 %) for RCP6.0, and 7%(18 %) for RCP8.5. The differences between changes in the summer surface ozone over the MB and changes in the tropospheric $O_3$ burden reflects the

10 fact that the surface $O_3$ over the MB is mainly controlled by reductions in precursor emissions and the $NO_x$-limited regime over the MB. Figure 11 shows the surface $O_3$ change between 2030 and 2000 (REF), 2100 and 2030, and 2100 and 2000. The ACCMIP models ensemble mean differences and their standard deviation are calculated for the period 2000-2100 over the Mediterranean region (see Fig. 2) and for the four RCPs. We use the Student t test for 95% confidence level to evaluate the statistical significance of surface $O_3$ changes over the Mediterranean region. For RCP2.6, the surface $O_3$ mean decreases

between 2000 and 2030 over the Mediterranean region (-5 ppbv), with a significant minimum in southern Europe mainly in Italy (-11 ppbv). An increase is observed in the northwest of Europe (+2 ppbv). However, over the period 2030-2100, the surface $O_3$ decreases significantly over the Mediterranean region (-11 ppbv) and specifically over the Mediterranean Sea and the eastern part of the Atlantic Ocean (-17 ppbv). Over the period 2000-2100, the surface $O_3$ decreases significantly on average by -15 ppbv. For RCP4.5, from 2000 to 2030, the ozone decrease is restricted to Europe and the Mediterranean Sea with an ozone

increase over North Africa and the eastern part of the Atlantic Ocean, reaching a maximum of +4 ppbv unlike the RCP2.6. Surface $O_3$ remains generally constant over the Mediterranean area (-1 ppbv). However, a significant reduction in ozone occurs between 2030 and 2100 over the Mediterranean region (-8 ppbv) and specifically over the Mediterranean Sea and the Middle East (-15 ppbv). For the RCP6.0 as for the RCP2.6, the surface $O_3$ decreases drastically over the Mediterranean region between 2000 and 2100 reaching -25 ppbv over the Mediterranean Sea. Despite the large radiative effect that characterizes the RCP6.0

scenario, we observe a net decrease in the surface $O_3$ concentration as for the RCP2.6 and even more pronounced than the RCP4.5. For the three scenarios, the surface $O_3$ change is likely due to the decrease in ozone precursors ($NO_x$, $CH_4$ and $CO$) and also to the $NO_x$-limited regime over the MB that connects ozone and its precursors. This means that $O_3$ decreases with NOx emission reductions. Water vapor is also one of the most important climate variables affecting tropospheric ozone (Jacob and Winner, 2009). High values of specific humidity are simulated over the Mediterranean Sea due to the evaporation (not

shown). That can explain the largest decrease in surface $O_3$ over the Mediterranean Sea and the eastern part of the Atlantic Ocean. The RCP8.5 is the only scenario that shows a very strong increase in $CH_4$, temperature and specific humidity as seen previously in Figs 8 and 9. These changes can be interpreted as a consequence of an intense climate change, despite the emission reduction policy and the chemical regime that promote the decrease in surface $O_3$. From 2000 to 2030, surface $O_3$ generally increases over the Mediterranean region (+2 ppbv) with a strong increase over the Arabian Peninsula (+8 ppbv)



and a local decrease in southern Europe reaching -5 ppbv. The trend in surface $O_3$ is opposite from +8 to -2 ppbv between 2030 and 2100 over the Middle East. The surface $O_3$ increases between 2000 and 2100, except in southern Europe and the northern part of the Mediterranean Sea. Note that the ACCMIP mean change in surface $O_3$ between 2030 and 2100 shows a marked East-West gradient with an increase in the West and a decrease in the East. This East-West gradient is represented by most of individual models (not shown). However, all the changes in surface $O_3$ are not significant at the 95% confidence level for the RCP8.5. CMAM is the only model that shows an increase over the entire Mediterranean region between 2000 and 2100. Note that CMAM is the model that is the closest to the observations with low $O_3$ concentrations compared to the other models (Fig. 7). The inter-model standard deviation (sd) between 2030 and 2100 (Fig. 11 bottom) is generally small with $\mathrm{sd} < 6\,\mathrm{ppbv}$ for the four scenarios, except for RCP2.6 over the Ligurian Sea ($\mathrm{sd} > 10\,\mathrm{ppbv}$), where some models provide a high concentration of $O_3$ (e.g. E2R-GISS). The disagreement between models over this region is highlighted in section 3.2.

In conclusion, we showed that surface $O_3$ decreases between 2000 and 2100 for RCP2.6, RCP4.5, RCP6.0 and that the relative changes for the surface $O_3$ over the MB decrease much more than the relative changes for the tropospheric ozone burden. For the RCP8.5, the surface $O_3$ remains constant between 2000 and 2100 over the MB. The decrease in surface $O_3$ is more pronounced for RCP2.6 and RCP6.0 than that for RCP4.5, which is mainly due to the reduction of ozone precursors. The largest decrease is observed over the Mediterranean Sea and the eastern part of the Atlantic Ocean. For the RCP8.5, the ACCMIP mean change in surface $O_3$ between 2030 and 2100 shows a marked East-West gradient with an increase in the West and a decrease in the East, but these changes are not statistically significant.

## 4.4 Effects of climate change and ozone precursors on future surface ozone changes

The future climate change is expected to influence the evolution of surface $O_3$ through changes in temperature, solar radiation and water vapor (Meleux et al., 2007; Forkel and Knoche, 2007; Hedegaard et al., 2008; Jacob and Winner, 2009; Katragkou et al., 2011; Lei et al., 2012; Hedegaard et al., 2013; Doherty et al., 2013), as well as the enhanced stratospheric contribution to surface ozone as a result of the increased Brewer Dobson circulation (Butchart and Scaife, 2001; Collins et al., 2003; Kawase et al., 2011; Lacressonnière et al., 2014). The impact of these climatic processes can be more marked on the MB which is a highly dynamical region (Lelieveld et al., 2002, 2009). The surface ozone changes are also controlled by changes in $O_3$ precursor emissions and methane concentration. Several studies have highlighted the importance of $CH_4$ emission control on surface $O_3$ (Fiore et al., 2008; Wild et al., 2012). Figure 12 shows the surface $O_3$ over the period from 2000 to 2100 as a function of the evolution of $NO_x$ emissions (Fig. 12a) and $CH_4$ concentration (Fig. 12b). The relationship between $O_3$ and $NO_x$ (Fig. 12a) is quasi-linear for RCP2.6, RCP4.5 and RCP6.0. A small decrease in $NO_x$ emissions implies a small decline in surface $O_3$ as for RCP4.5 and a large decrease in $NO_x$ leads to a more pronounced decrease in $O_3$ as for RCP2.6 and RCP6.0. Young et al. (2013) show the same linear relationship by comparing the $NO_x$ emissions and the global modeled tropospheric $O_3$ burdens, but with a smaller decrease in tropospheric $O_3$ as seen in section 4.3. For RCP8.5, the linear relationship between the two variables ($NO_x$ emissions and surface $O_3$) is no longer valid. Despite the decrease in $NO_x$ emissions, surface $O_3$ increases slightly for 2030 as for 2100. The changes in $CH_4$ concentration has no apparent impact on the changes in surface $O_3$ for RCP2.6, RCP4.5 and RCP6.0 (Fig. 12b). Whether the $CH_4$ concentration decreases (RCP2.6) or remains constant (RCP6.0),





the surface $O_3$ decline is similar in magnitude. The RCP8.5 is marked by a nearly double increase in $CH_4$ concentration, which is associated with a slight increase in surface $O_3$. This shows that a positive correlation between changes in $CH_4$ and surface $O_3$ can exist. Therefore, it can be deduced for the RCP8.5 that a warmer climate associated with a strong increase in $CH_4$ concentration will offset the benefit of the emission reductions. This effect of climate change called "climate penalty" (Wu et al., 2008) was already highlighted over Europe by several studies and summarized by Colette et al. (2015). Wild et al. (2012) show that 75% of the average difference (5 ppbv) in surface $O_3$ between the outlying RCP2.6 and RCP8.5 scenarios can be attributed to differences in $CH_4$ abundance. We note that, for the RCP8.5, the change in surface $O_3$ over the MB is less intense than the global tropospheric $O_3$ change highlighted by Young et al. (2013).

The implementation of different RCPs was done by independent modeling groups and they are based on different RF levels that were chosen from the literature. This makes the interpretation of our results regarding the different RCPs more complex. Nevertheless, the comparison of scenarios can then be used to give a partial interpretation of the effect of climate change and $CH_4$ changes on surface $O_3$ evolution. The magnitudes of the changes in temperature, specific humidity and $CH_4$ concentrations are different for RCP2.6, RCP4.5 and RCP6.0. We note that the $O_3$ evolutions are almost the same for these scenarios over the period 2000-2100, despite the marked difference in the global RF of 3.4 $W.m^{-2}$ which is mainly dominated by the forcing from $CO_2$ (Van Vuuren et al., 2011). The RCP6.0 can be considered as a scenario that could significantly decrease the future surface $O_3$ over the MB. The beneficial effects of climate change through the increase of specific humidity (Jacob and Winner, 2009) and the reduction policy of ozone precursors play an important role for the changes in surface $O_3$. The RCP8.5 is atypical and different from the other scenarios. The surface $O_3$ over the MB remains constant over the period 2000-2100 with a strong increase in temperature, specific humidity and $CH_4$ concentration, unlike the global tropospheric ozone, which should increase by 18% in 2100 (Young et al., 2013).

In conclusion, surface $O_3$ decreases over the MB for the RCP2.6, RCP4.5 and RCP6.0 mainly due to the reduction policy of $O_3$ precursors associated with the $NO_x$-limited regime combined to a beneficial effect of climate change through the increase of specific humidity over the MB. For the RCP8.5, the future climate change associated with a net increase in $CH_4$ concentration offsets the benefit of the emission reductions. We estimate that the MB will benefit from both the $CH_4$ and $NO_x$ emissions control.

### 4.5 Future changes in ozone budget

In this section, we focus on the evolution of the ozone budget along the $21^{st}$ century over the MB: production (P), chemical loss (L), production minus chemical loss (P-L) and dry deposition of $O_3$ (D) for all scenarios and periods. Figure 13 shows the relative changes in summer surface $O_3$ budget terms (P, L, P-L and D) over the MB for the RCP2.6, RCP4.5 and RCP6.0. In terms of the chemical ozone budget evolution, we observe that all the terms P, L and P-L decrease for RCP2.6 by 2030 and 2100 compared to REF, although the percentage decrease is almost the same for P and L by 2100, which explains the similar decrease of -40% in the P-L term. For the RCP4.5 and RCP6.0, all the terms decrease by 2100 after a slight increase in P-L (11%) by 2030. We note that all models are in agreement in terms of trends between 2030 and 2100 for these three scenarios. For the RCP8.5 scenario (Fig. 14), the averages of P, L and P-L increase by 2030. From 2030 to 2100 time slice, the mean





relative changes of P, L and P-L are -10%, 5% and -15%, respectively. Nevertheless, the models are not in agreement in terms of the chemical ozone budget evolution for the RCP8.5. The terms (P, L and P-L) decrease for GFDL-AM3, STOC-HadAM3 and UM-CAM and increase for CMAM and CESM-CAM-superfast. It is difficult to interpret this difference given the complexity of the models. Nevertheless, we note that the models closest to the observations (section 3) are those with increasing chemical

5    terms and conversely for the models that overestimate surface $O_3$. Lacressonnière et al. (2014) have shown that the term P-L decreases over Europe in the short-term period (2030 and 2050) using the MOCAGE chemical transport model for RCP8.5, and Young et al. (2013) have also shown that the net chemical production (P-L) of the global tropospheric $O_3$ decreases between the REF period and 2100 for the RCP8.5. Also note that each scenario is represented by a different set of models, except for RCP2.6 and RCP8.5 that are represented by the same set of models, making them comparable in terms of future $O_3$ budget

10   trend. The net influx of ozone is not investigated due to its large uncertainty within a MB box and the limited amount of ACCMIP data (Young et al., 2013). Dry deposition of $O_3$ decreases for all scenarios from 2030 to 2100, in a proportional way to that of the surface $O_3$. Moreover, the surface $O_3$ budget terms (P, L, P-L and D) decrease by 2100 over the MB for the RCP2.6, RCP4.5 and RCP6.0, with a general agreement between models. For the RCP8.5, the models are not consistent for the surface $O_3$ budget terms evolution, which explains the non-significant changes in surface ozone over the MB.



## 5   Conclusions

The future evolution in surface ozone is investigated in summertime (June, July and August) over the Mediterranean basin (MB), from 2000 to 2100 using 11 chemistry models that have contributed to the Atmospheric Chemistry and Climate Model Intercomparison Project (ACCMIP). This study was carried out over the MB, considering time slices around 2000, 2030 and 2100, and using the four Representative Concentration Pathways (RCPs). We started by assessing the models used by comparing surface ozone between contemporary era ACCMIP simulations (1990-2010) and gridded observations from the EMEP, WMO-GAW and Airbase network over the MB. Our approach consists firstly of studying the meteorological parameters (temperature, specific humidity, precipitation) and ozone precursors ($CH_4$ concentration, $NO_x$, VOCs, CO emissions). In a second step, we analyzed the changes in surface ozone and available terms of its budget (chemical budget and dry deposition). The evaluation of the models against observations over a REF period (2000 time slice) allowed us to understand their behavior to simulate surface ozone. The annual cycle is very well captured by most of the models and the ACCMIP mean shows better performances than most of the models with a correlation coefficient R = 0.93. However, we found that most models overestimate the summer surface observations with $O_3$ being better represented in southern Europe than in the Mediterranean Sea and with a modeled spatio-temporal variability smaller than the observed variability over the MB. The model performances do not change between the global and the regional scales. The analysis of meteorological parameters indicates that the temperature increases during the $21^{st}$ century for all RCPs, according to the radiative forcing (RF), by an average of 5.4 K in 2100 compared to 2000. The specific humidity increases also as a response to the rise of the temperature, precipitation decreases for scenarios that have high RF (RCP6.0 and RCP8.5). Changes in ozone precursors show that CO and $NO_x$ decrease constantly, reflecting the emission reduction policy. Changes in $O_3$ concentrations due to VOCs emissions changes are not conclusive given the very large inter-model variability in biogenic VOCs emissions. $CH_4$ drastically increases for RCP8.5 but decreases for other scenarios. The RCP8.5 shows a non-significant change in summer surface ozone of -0.4% (0.2%) in 2030 (2100) over the MB, unlike the other RCPs, which show an $O_3$ decrease of -13% (-36%) for RCP2.6, -7% (-22%) for RCP4.5 and -11% (-33%) for the RCP6.0. The difference between changes in the summer surface $O_3$ over the MB and changes in the tropospheric $O_3$ burden reflects the fact that the surface $O_3$ over the MB is more controlled by reductions of its precursor emissions, water vapor represented by the increase in the specific humidity and the $NO_x$-limited regime over the MB. The net chemical budget (chemical production minus loss) of ozone decreases intensively from 2030 to 2100 for RCP2.6, RCP4.5 and RCP6.0 and less strongly for the RCP8.5. Dry deposition of ozone decreases for all RCPs following surface $O_3$ concentration increases, especially for RCP2.6 and RCP6.0 that show a large ozone increase. The net decrease in surface ozone over the MB for RCP2.6, RCP4.5 and RCP6.0 is mainly due to the reduction in ozone precursors emissions. This reduction is the same for RCP2.6 and RCP6.0, despite the marked difference in the global RF of 3.4 $W.m^{-2}$ between the two scenarios, which is mainly dominated by the forcing from $CO_2$. The increased ozone for RCP8.5 over the Mediterranean region except the southern Europe, shows how the future climate change associated with a net increase in $CH_4$ concentrations can offset the benefit of the emission reductions. The largest decrease in surface ozone is calculated over the Mediterranean Sea and the Eastern part of the Atlantic Ocean, likely due to the increase of specific humidity in these areas. Other dynamical factors can affect the trends in surface ozone over the



MB (e.g. increasing stratosphere-troposphere exchange, the recovery of stratospheric ozone, long-range transport, etc.). Future modeling studies should quantify the sensitivity of the future surface $O_3$ to climate change and $CH_4$ concentrations changes over the MB.

*Acknowledgements.* This work is funded in France by the Centre National de Recherches Météorologiques (CNRM) of Météo-France, the
5   region Midi-Pyrénées and the Centre National de Recherches Scientifiques (CNRS).



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





**Figure 1.** List of the ACCMIP model used in this study and the time-slice availability for each model. Gray colors stands for model outputs not available and blue color stands for available model outputs.





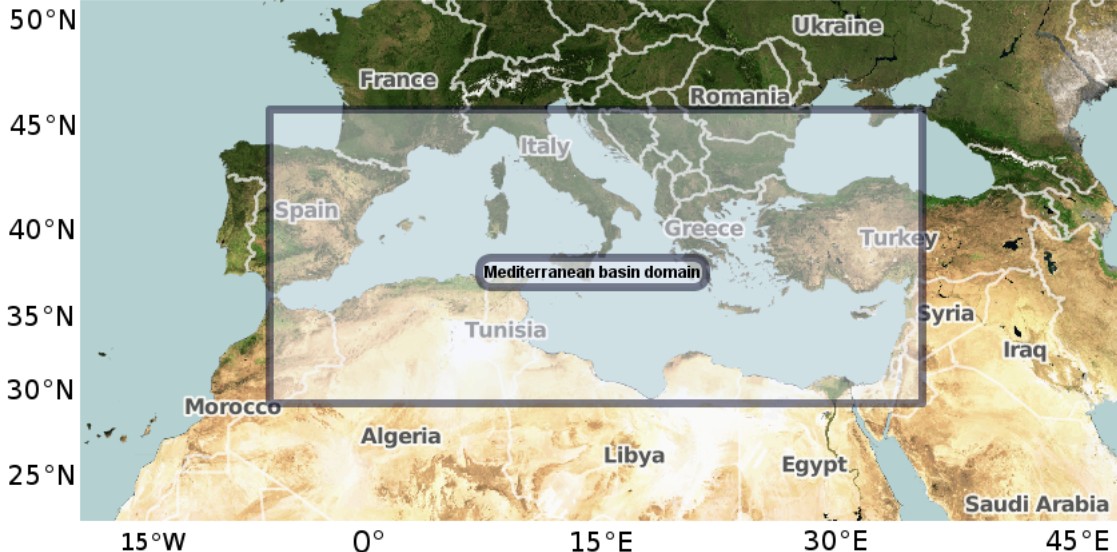

**Figure 2.** The Mediterranean region including southern Europe, northern Africa and a part of the Middle East. The gray box represents the Mediterranean Basin (MB) domain, in which the statistical analysis is performed.





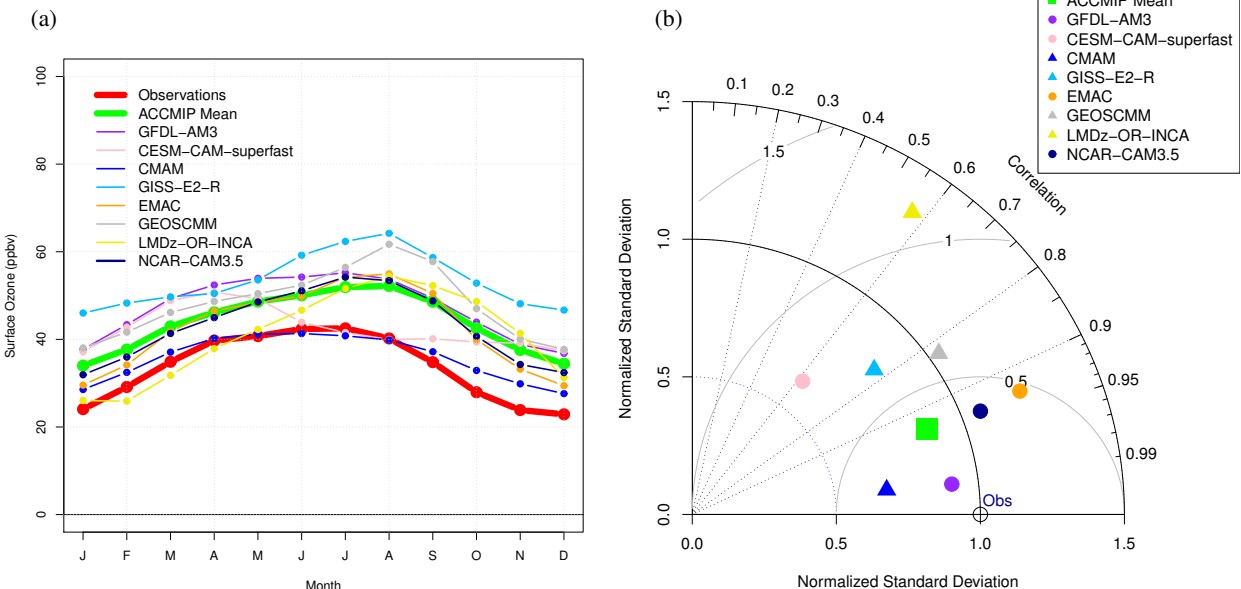

**Figure 3.** (a) Annual cycle of surface ozone from ACCMIP models averaged over the period 1990-2010 and over the Mediterranean basin (thin line), between gridded observations (thick red line), ACCMIP ensemble mean (thick green line) and the ACCMIP ensemble. (b) Taylor diagram of the annual cycle of surface ozone averaged over the period 1990-2010. The radial coordinate shows the standard deviation, normalized by the observed standard deviation. The azimuthal variable shows the correlation of the modeled annual cycle with the observed annual cycle. The normalized root mean square error is indicated by the grey circle centered on the observational reference (Obs) point. Obs is indicated by the open circle on the x-axis. The analysis is performed over the Mediterranean Basin domain (see Fig. 2).



(a)                                                    (b)

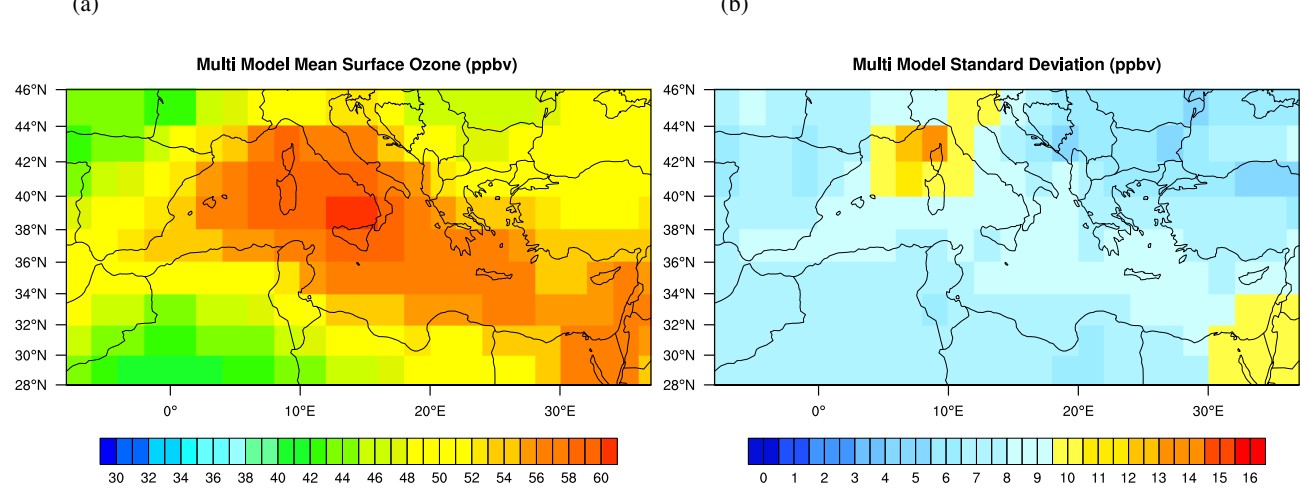

**Figure 4.** ACCMIP ensemble mean of surface ozone concentration in ppbv (a) and the ACCMIP ensemble standard deviation in ppbv (b) over the REF period (2000 time slice) from the historical experiment over the Mediterranean basin.





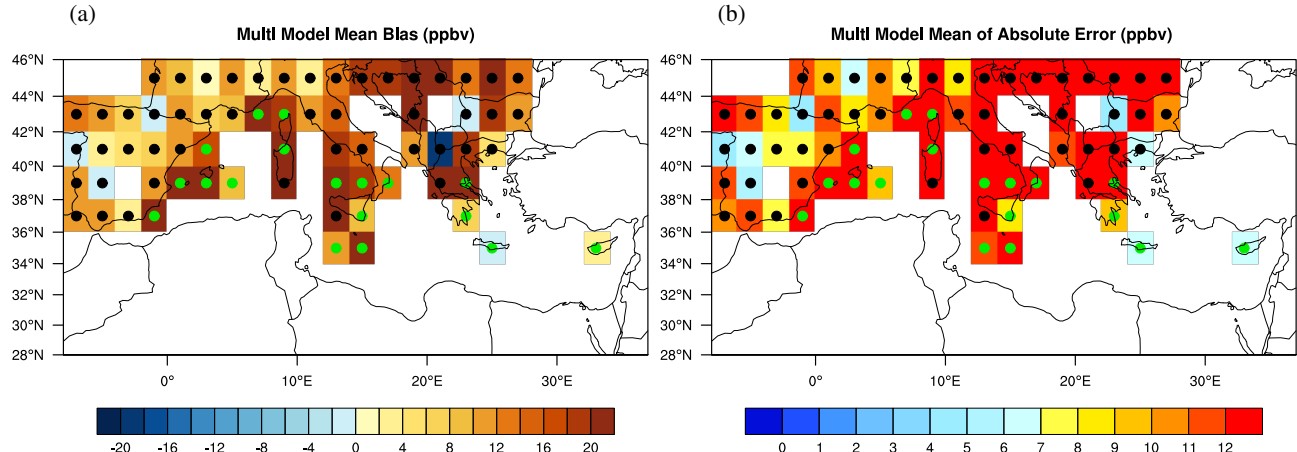

**Figure 5.** ACCMIP ensemble mean bias of surface ozone concentration in ppbv (a) and the ACCMIP ensemble mean error in ppbv (b) over the REF period (2000 time slice) from the Hist experiment over the Mediterranean basin. Black and green filled circles represent land and sea, respectively.





**Figure 6.** Summer (JJA) surface ozone (ppbv) over the period 1990-2010. Comparison between observations (x axis) and ACCMIP model results (y axis). The sea data points are colored in cyan whereas the land data points are colored in red. Each dot represents a grid-point per summer (JJA) of year. The normalized mean biases (NMB) and the Coefficient of variation ratio (CvR) are given in the inset. The diagonal reference 1-1 line (dashed) is also shown. The analysis is performed over the Mediterranean Basin domain (see Fig. 2).





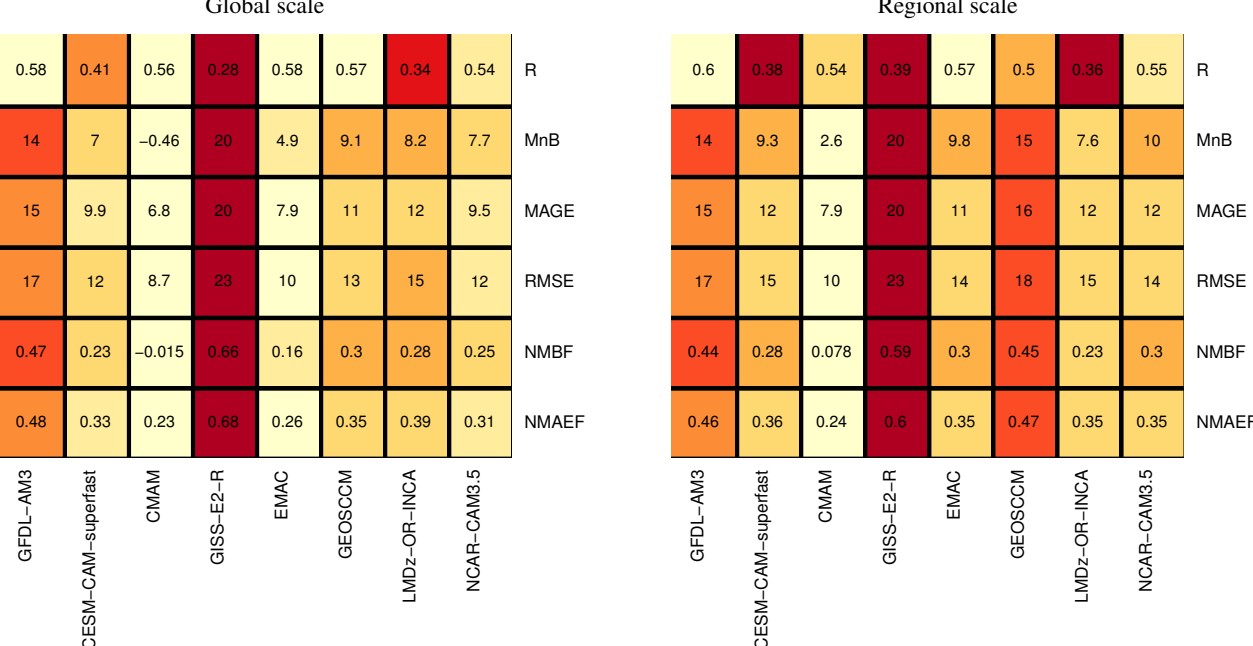

**Figure 7.** ACCMIP model performances, based on spatio-temporal comparison of summer surface ozone between observations and ACCMIP models computed over a REF period (1990-2010) from the Hist experiment. Row and columns represent individual metrics and models, respectively. Each cell contains the value of a corresponding metric and a color indicating the performance of the model, from white (close to the observations) to red (far from the observations). The used metrics are: correlation coefficient (R), mean bias (MnB), mean absolute gross error (MAGE), root mean square error (RMSE), the normalized mean bias factor (NMBF) and the normalized mean absolute error factor (NMAEF). Each metric is calculated at global (left) and regional scales (right).





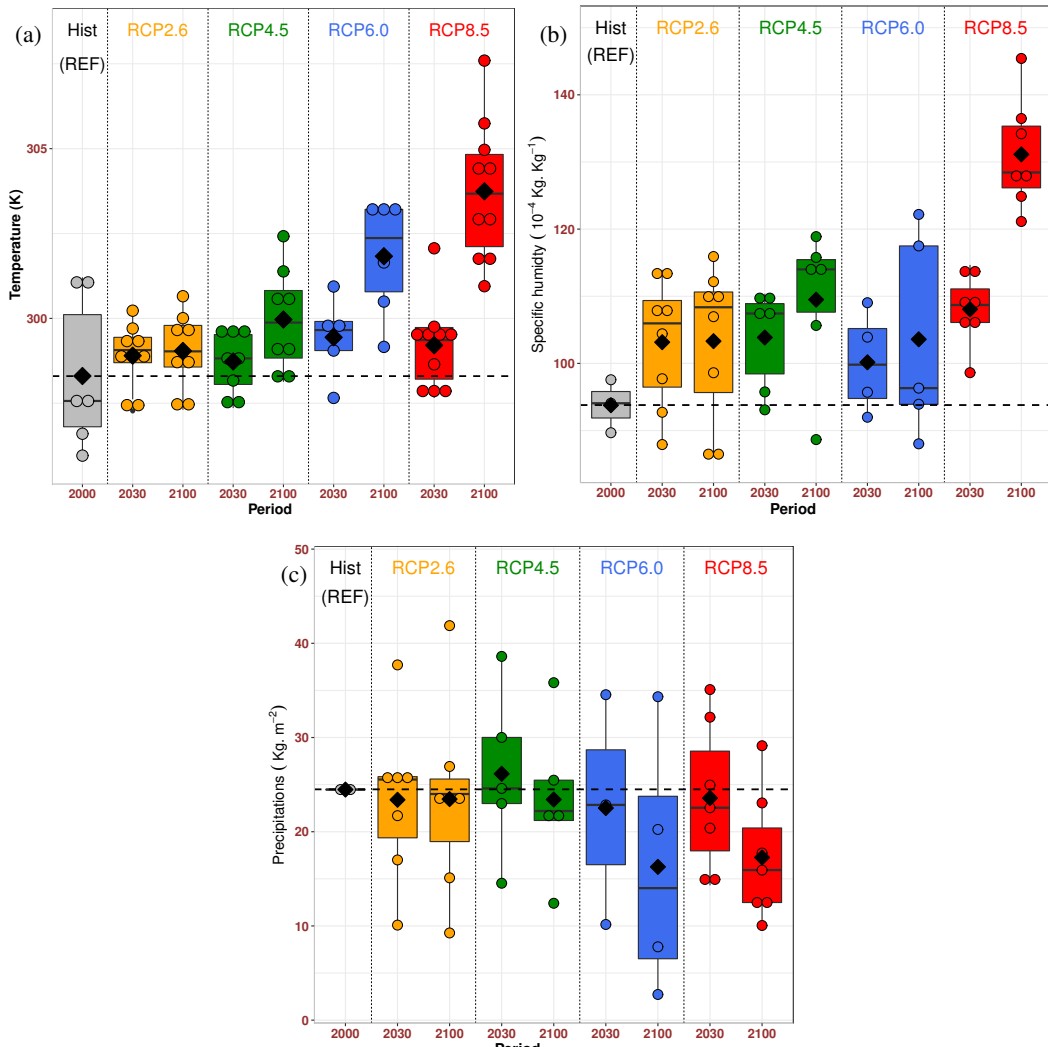

**Figure 8.** Box-whisker plots of the annual average of (a) temperature in kelvin (K), (b) specific humidity and (c) precipitation since 2000, calculated over the Mediterranean Basin domain (see Fig. 2) for the JJA period and for the four Representative Concentration Pathways. The median is indicated by the thick horizontal black line, the multi model mean by a filled diamond, the (25-75%) range by the colored box and minimum/maximum excluding outliers by whisker. Each filled circle represents a single model.





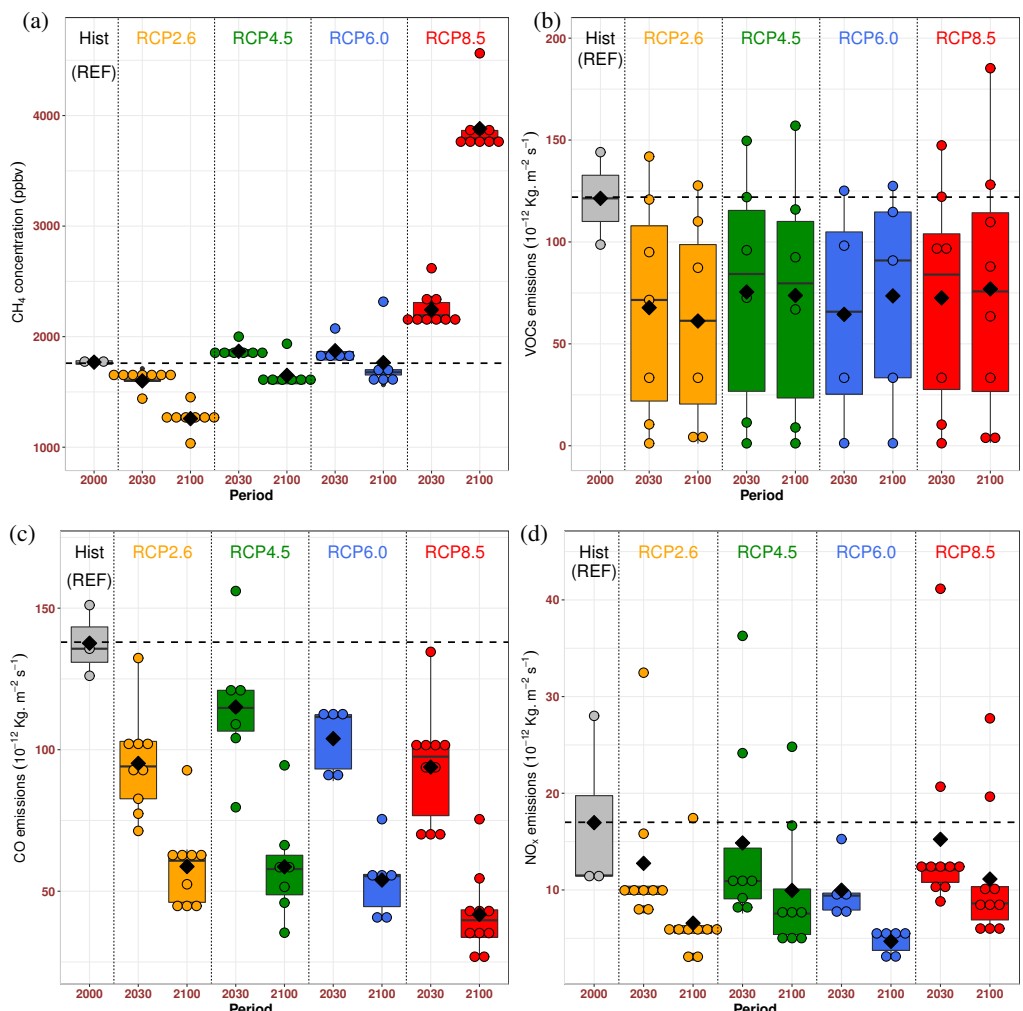

**Figure 9.** Same as Fig. 8 but for the surface $CH_4$ concentrations (a) and emissions of ozone precursors: VOCs (b), CO (c) and $NO_x$ (d).





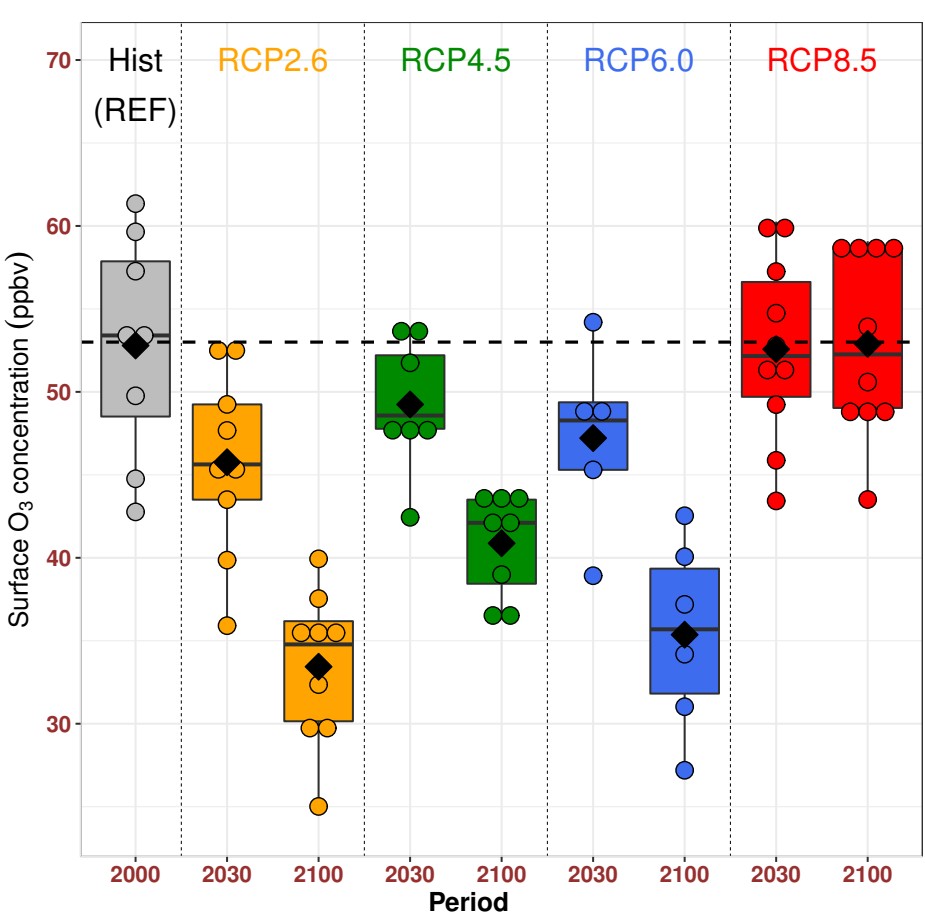

**Figure 10.** Same as Fig. 8 but for the surface $O_3$ concentrations (ppbv).



**Figure 11.** ACCMIP ensemble mean surface ozone change (1st–3rd row) and standard deviation in ppbv (4th row) between 2000 and 2100. Each column represents a Representative Concentration Pathways scenario. The rows from top to bottom correspond to anomalies in surface ozone concentration : 2030-REF, 2100-REF, 2100-2030, 2100-REF and the standard deviation of anomalies 2100-2030, respectively. REF represents the 2000 situation. Black dots indicate regions in the maps with non-significant changes at the 95% confidence level.




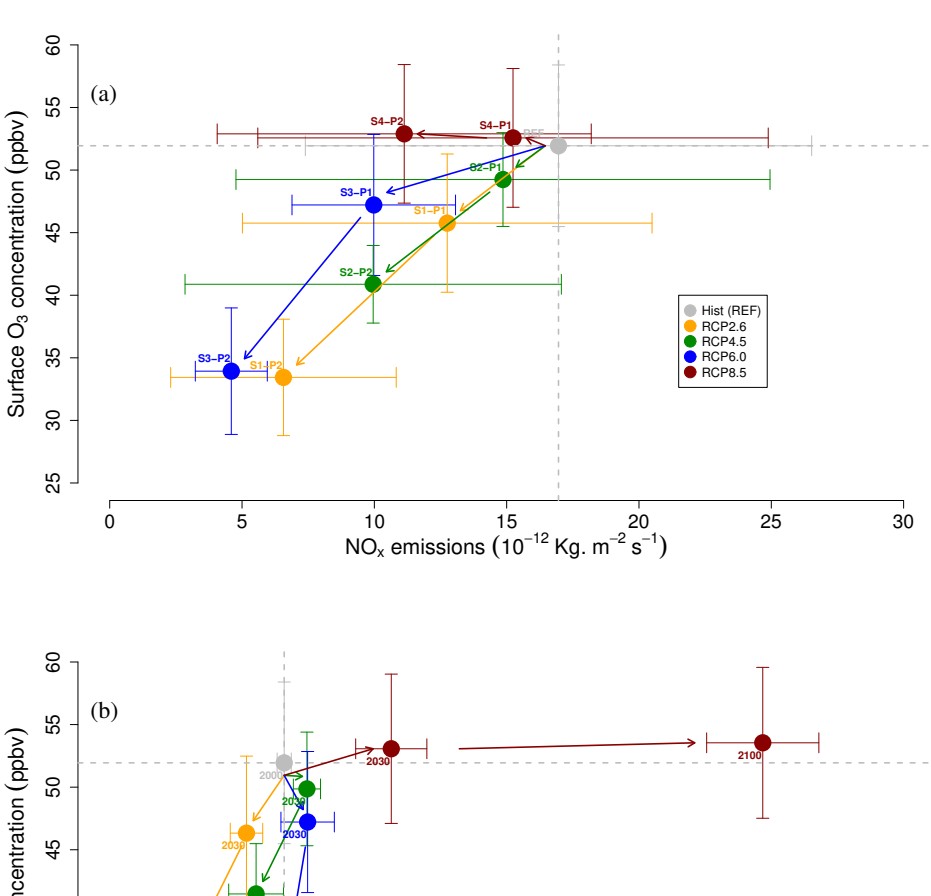

**Figure 12.** ACCMIP model ensemble mean change in the surface ozone (ppbv) as a function of (a) changes in total $NO_x$ emissions $(10^{-12} Kg.m^{-2}.s^{-1})$ and (b) changes in the surface $CH_4$ concentration (ppbv), calculated over the Mediterranean Basin domain (see Fig. 2) for the JJA period and for the four Representative Concentration Pathways inset box. Error bars indicate multi-model standard deviation. Dashed lines refer to REF values (2000 time slice).

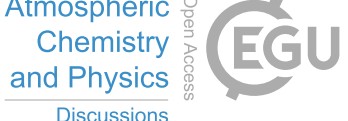



**Figure 13.** Future relative change in surface ozone budget over the MB domain, (a) chemical production (P), (b) chemical loss (L) and (c) chemical budget (P-L) of surface ozone, (d) dry deposition of ozone (D). calculated over the Mediterranean Basin for JJA period and for the RCP2.6, RCP4.5 and RCP6.0. The median is indicated by the thick horizontal black line, the multi models mean by a filled diamond, the (25-75%) range by the colored box and minimum/maximum excluding outliers by whisker. Each colored point represents a single model. The dashed horizontal line represents the mean for the REF period (2000) and considered as a reference.





**Figure 14.** Same as Fig. 13 but for the RCP8.5.



**Table 1.** List of the historical ACCMIP simulations (2000 time slice) used in this study and the availability of data for each model and parameter.

(★) = available, (-) = not available.

| Model | Temperature | Specific humidity | Precipitation | CH$_4$ | NO$_x$ | VOCs | CO | O$_3$ |
|---|---|---|---|---|---|---|---|---|
| CESM-CAM-Superfast | - | - | - | - | - | - | - | ★ |
| CMAM | ★ | ★ | ★ | ★ | ★ | - | ★ | ★ |
| EMAC-DLR | ★ | ★ | - | ★ | ★ | ★ | ★ | ★ |
| GEOSCCM | ★ | - | - | - | - | - | - | ★ |
| GFDL-AM3 | ★ | ★ | ★ | ★ | ★ | ★ | ★ | ★ |
| GISS-E2-R | ★ | - | - | - | - | - | - | ★ |
| HADGEM2 | - | - | - | - | - | - | - | - |
| LMDZORINCA | - | - | - | - | - | - | - | ★ |
| NCAR-CAM3.5 | - | - | - | - | - | - | - | ★ |
| STOC-HadAM3 | - | - | - | - | - | - | - | - |
| UM-CAM | - | - | - | - | - | - | - | - |





**Table 2.** List of the future ACCMIP simulations used in this study and the availability of data for each model, parameter and scenario. (★) = available for the periods 2030 and 2100 except for GEOSCCM which is available only in 2100, (-) = not available.

| Model | Scenario | Temperature | Specific humidity | Precipitation | CH$_4$ | NO$_x$ | VOCs | CO | O$_3$ |
|---|---|---|---|---|---|---|---|---|---|
| CESM-CAM-Superfast | RCP2.6 | ★ | ★ | ★ | ★ | ★ | ★ | ★ | ★ |
| | RCP6.0 | ★ | ★ | ★ | ★ | ★ | ★ | ★ | ★ |
| | RCP8.5 | ★ | ★ | ★ | ★ | ★ | ★ | ★ | ★ |
| CMAM | RCP2.6 | ★ | ★ | ★ | ★ | ★ | - | ★ | ★ |
| | RCP4.5 | ★ | ★ | ★ | ★ | ★ | - | ★ | ★ |
| | RCP8.5 | ★ | ★ | ★ | ★ | ★ | - | ★ | ★ |
| EMAC-DLR | RCP4.5 | ★ | - | - | ★ | ★ | ★ | ★ | ★ |
| | RCP8.5 | ★ | - | - | ★ | ★ | ★ | ★ | ★ |
| GEOSCCM | RCP6.0 | ★ | ★ | ★ | ★ | ★ | ★ | ★ | ★ |
| GFDL-AM3 | RCP2.6 | ★ | ★ | ★ | ★ | ★ | ★ | ★ | ★ |
| | RCP4.5 | ★ | ★ | ★ | ★ | ★ | ★ | ★ | ★ |
| | RCP6.0 | ★ | ★ | ★ | ★ | ★ | ★ | ★ | ★ |
| | RCP8.5 | ★ | ★ | ★ | ★ | ★ | ★ | ★ | ★ |
| GISS-E2-R | RCP2.6 | ★ | ★ | ★ | ★ | ★ | ★ | ★ | ★ |
| | RCP4.5 | ★ | ★ | ★ | ★ | ★ | ★ | ★ | ★ |
| | RCP6.0 | ★ | ★ | ★ | ★ | ★ | ★ | ★ | ★ |
| | RCP8.5 | ★ | ★ | ★ | ★ | ★ | ★ | ★ | ★ |
| HADGEM2 | RCP2.6 | ★ | ★ | ★ | ★ | ★ | ★ | ★ | ★ |
| | RCP4.5 | ★ | ★ | ★ | ★ | ★ | ★ | ★ | ★ |
| | RCP8.5 | ★ | ★ | ★ | ★ | ★ | ★ | ★ | ★ |
| LMDZORINCA | RCP2.6 | ★ | - | - | ★ | ★ | ★ | ★ | ★ |
| | RCP4.5 | ★ | - | - | ★ | ★ | ★ | - | ★ |
| | RCP6.0 | ★ | - | - | ★ | ★ | ★ | ★ | ★ |
| | RCP8.5 | ★ | - | - | ★ | ★ | ★ | ★ | ★ |
| NCAR-CAM3.5 | RCP2.6 | ★ | ★ | - | ★ | ★ | - | ★ | ★ |
| | RCP4.5 | ★ | ★ | - | ★ | ★ | - | ★ | ★ |
| | RCP6.0 | ★ | ★ | - | ★ | ★ | - | ★ | ★ |
| | RCP8.5 | ★ | - | - | ★ | ★ | - | ★ | ★ |
| STOC-HadAM3 | RCP2.6 | ★ | ★ | ★ | ★ | ★ | ★ | ★ | ★ |
| | RCP8.5 | ★ | ★ | ★ | ★ | ★ | ★ | ★ | ★ |
| UM-CAM | RCP2.6 | ★ | ★ | ★ | - | ★ | ★ | ★ | ★ |
| | RCP4.5 | ★ | ★ | ★ | - | ★ | ★ | ★ | ★ |
| | RCP8.5 | ★ | ★ | ★ | - | ★ | ★ | ★ | ★ |





**Table 3.** Definition of the metrics used to evaluate the ACCMIP model performances. O and M refer to observations and model, respectively. $\bar{O} = \frac{1}{N} \sum_{i=1}^{n} O_i$, $\bar{M} = \frac{1}{N} \sum_{i=1}^{n} M_i$.

| Metrics | Mathematical expression | Range |
|---|---|---|
| Normalized mean bias | $\mathrm{NMB} = \frac{\sum_{i=1}^{n}(M_i - O_i)}{\sum_{i=1}^{n} O_i}$ | $-1$ to $+\infty$ |
| Coefficient variation ratio | $\mathrm{CvR} = \frac{\frac{\sqrt{\frac{1}{N}\sum_{i=1}^{n}(O_i - \bar{O})^2}}{\bar{O}}}{\frac{\sqrt{\frac{1}{N}\sum_{i=1}^{n}(M_i - \bar{M})^2}}{\bar{M}}}$ | $0$ to $+\infty$ |
| Mean Bias | $\mathrm{MnB} = \frac{1}{N}\sum_{i=1}^{n}(M_i - O_i) = \bar{M} - \bar{O}$ | $-\bar{O}$ to $+\infty$ |
| Correlation coefficient | $R = \frac{\sum_{i=1}^{n}(M_i - \bar{M})(O_i - \bar{O})}{\left\{\sum_{i=1}^{n}(M_i - \bar{M})^2 \sum_{i=1}^{n}(O_i - \bar{O})^2\right\}^{\frac{1}{2}}}$ | $-1$ to $+1$ |
| Root mean square error | $\mathrm{RMSE} = \sqrt{\frac{1}{N}\sum_{i=1}^{n}(M_i - O_i)^2}$ | $0$ to $+\infty$ |
| Mean absolute gross error | $\mathrm{MAGE} = \frac{1}{N}\sum_{i=1}^{n}|(M_i - O_i)|$ | $0$ to $+\infty$ |
| Normalized mean bias factor | $\mathrm{NMBF}(\bar{M} \geq \bar{O}) = \frac{\sum(M_i - O_i)}{\sum O_i}$ $\mathrm{NMBF}(\bar{M} < \bar{O}) = \frac{\sum(M_i - O_i)}{\sum M_i}$ | $-\infty$ to $+\infty$ |
| Normalized mean absolute error factor | $\mathrm{NMAEF}(\bar{M} \geq \bar{O}) = \frac{\sum|M_i - O_i|}{\sum O_i}$ $\mathrm{NMAEF}(\bar{M} < \bar{O}) = \frac{\sum|M_i - O_i|}{\sum M_i}$ | $0$ to $+\infty$ |