# Peer review of "Future changes in surface ozone over the Mediterranean basin in the framework of the Chemistry-Aerosol Mediterranean Experiment (ChArMEx)"

_Atmospheric Chemistry and Physics, 2017_

## Referee Comment (RC1) · Anonymous Referee #1 · 20 Aug 2017

MS No.: acp-2017-553 Future surface ozone over the Mediterranean Basin using the ACCMIP outputs of 11 models as predicted for 2000, 2030 and 2100 and based on four representative concentration pathways was carried out. Each of these four pathways results are compared to measure surface ozone datasets. This study serves as an important contribution to the ChArMEx Project and as such, deserves its publication in ACP. However, few of the interpretations given by the authors explaining the discrepancies obtained between the simulated concentrations as compared to the present and future reference periods are expressed in a too hypothetical manner and ought to be

based on a more scientific basis based on further references of similar studies. Consequently, I support its publication after treating the following comments. Major comments: 1) L205-215: The results obtained by NMB and CVR models as regarded to the model's overestimate of surface O3 are clear. However, some more explanations are necessary (e.g., sensitivity analyses to VOCs emissions) 2) L195: Please discuss the possible reasons for the model's overestimating surface O3 3) L235: Please discuss the possible reasons for the best performance obtained by CMAM while comparing surface O3 over the Mediterranean Basin and ACCMIP simulations. The rational for referring to Parrish et al. (2014) in this context is not clear. 4) L298: I don't see the point in discussing the ACCMIP inter-model variability of total VOC species since, as the authors mentioned, biogenic emissions are not specified and the VOC module differs from one model to another. The contribution of biogenic VOC (BVOC) emissions to regional ozone formation is significant. Over most of world's regions, the leading BVOC species, isoprene, reacts rapidly with hydroxyl radicals to form ozone (Pierce et al., 1998). The high ratio of BVOC to total VOC emission makes BVOC emissions a significant contributor to regional ozone concentrations. Estimations of BVOC emissions are subject to large uncertainty and are not well estimated by proxy concentrations (i.e. formaldehyde) observed from space. Hence, estimates of isoprene emissions by different biogenic emission models can vary more than a factor of two (Shim et al., 2005). 5) L182-188: The explanations for the high SD values over the Ligurian Sea and around Marseille are not clear. For example, do Vautard et al. (2007) give some evidences indicating that the stagnating conditions are the reason? What about the high SDs over the south-eastern parts of the Mediterranean Basin? 6) L160: I wonder, whether is not it obvious that a comparison of models to surface O3 averages for such long periods (8 to 16 years and for 1990-2010) is expected to give relative elevated correlations?

References to Major comments: T. Pierce, C. Geron, L. Bender, R. Dennis, G. Tonnesen, A. Guenther: Influence of increased isoprene emissions on regional ozone modeling, J. Geophys. Res. Atmos., 103 (D19) (1998), pp. 25611–25629. C. Shim, Y. Wang, Y. Choi, P.I. Palmer, D.S. Abbot, K. Chance: Constraining global isoprene emissions with Global Ozone Monitoring Experiment (GOME) formaldehyde column measurements, J. Geophys. Res. Atmos., 110 (D24) (2005) https://doi.org/10.1029/2004JD005629.

Minor comments: 1) L153-154: The maximum observed concentration recorded in July as compared to the ACCMP models August average for 1990-2010 contradicts the statement made by the authors in line 154. 2) L175: Please replace "Northern Africa" by south-east Mediterranean Basin. 3) L377: Please supply further explanations for the term "highly dynamical". 4) L389: The sentence: "..This shows that a positive correlation between changes in CH4 and surface O3 can exist" is not clear, please elaborate.

---

## Referee Comment (RC2) · Anonymous Referee #2 · 5 Sep 2017

This is a paper of interest to the CHARMEX community, providing a reasonably standard analysis of a series of models and their performance over the Mediterranean basin for various scenarios (RCP; time slices). However, I find that the paper describes many model results, but does not attempt sufficiently to explain their performance (e.g., why they differ from observations, why there are outliers). Furthermore, I see little evidence explaining the credibility of the model results for the future scenarios. For this reason, in my view the paper is not ready for publication in ACP. The authors should try to

address these points, perhaps engaging in intelligent speculation if needed.

The authors should also address the specific comments below.

Specific comments

P. 3:

L. 22: With multiple citations stick to either an alphabetical scheme or a chronological scheme, but do not mix them.

P. 6

L. 84: uses -> use.

P. 9

L. 158: Introduce acronym for ppbv. Make sure all acronyms are introduced, both in the abstract and in the main manuscript.

P. 10

L. 171: Why do these models represent best the annual cycle of surface O3?

P. 12

L. 214: Why do these models show less variability than the observations?

P. 13

L. 239-242: What is the reason for this behaviour in the models?

P. 14

L. 256: Is this trend significant?

L. 267: probably -> likely.

P. 15

[Figure]

L. 279: asset -> assets.

P. 16

L. 306: Why are these models outliers?

P. 17

L. 314: You mean the simulated emissions, correct?

P. 19

L. 364-370: How credible are these model results?

P. 23

Conclusions: The authors should consider splitting this into a discussion section and a short conclusions section.

Table 1: The paper mentions 11 models, but the table suggests that less are used. Could the authors clarify this.

Fig. 4 caption: I suggest you indicate what the endpoints of the colour scale mean. Same for similar figures.

Fig, 6 caption. Indicate what the coloured ellipses mean.

Fig. 7 caption: "... The metrics used...".

Fig. 8 caption: Identify the colour scheme in the caption. Same for Figs. 12, 13.

---

## Author Comment (AC1) · 4 Jan 2018

**Responses to reviewer 1**

**Future changes in surface ozone over the Mediterranean basin in the framework of the Chemistry-Aerosol Mediterranean Experiment (ChArMEx)**

Future surface ozone over the Mediterranean Basin using the ACCMIP outputs of 11 models as predicted for 2000, 2030 and 2100 and based on four representative concentration pathways was carried out. Each of these four pathways results are compared to measure surface ozone datasets. This study serves as an important contribution to the ChArMEx Project and as such, deserves its publication in ACP. However, few of the interpretations given by the authors explaining the discrepancies obtained between the simulated concentrations as compared to the present and future reference periods are expressed in a too hypothetical manner and ought to be based on a more scientific basis based on further references of similar studies.  Consequently, I support its publication after treating the following comments.

We thank the reviewer for his/her valuable comments. In this new version of manuscript, we present all available data (including outliers) in the reference period (1990-2010). In particular, our statistics include the results from MOCAGE and MIROC-CHEM, increasing the number of models from 11 to 13 models. The figures and the values are updated accordingly in this new version of paper. Below, we answer point to point to the Major and the Minor comments in blue:

**Major comments**

**1)** **L205-215:** The results obtained by NMB and CVR models as regarded to the model's overestimate of surface O3 are clear. However, some more explanations are necessary (e.g., sensitivity analyses to VOCs emissions)

The sensitivity analyses are not possible to perform with this dataset. We only used model outputs from simulations already performed by several different modeling groups in the framework of the ACCMIP inter-comparison project. Several studies have highlighted a high ozone bias in the Northern Hemisphere and a low ozone bias in the Southern Hemisphere (e.g. Young et al., 2013; Bowman et al., 2013; Travis et al., 2016). This bias appears systematically in most models, suggesting that a deficiency in the ozone precursor emissions can partly explain a bias in the outputs of the models. However, the variability of the modeled surface $O_3$ is smaller than that of the observations because the emissions inventory in the models does not change year to year over the period 1990-2010, unlike the surface ozone observations which fluctuate over this period. Consequently, we removed this section which provide a descriptive statement and does not bring

any added value to our paper. In addition, the outcomes from this section (section 3.3) concerning the ozone overestimation over the sea are now discussed in the previous section (section 3.2).

**2) L195:** Please discuss the possible reasons for the model's overestimating surface O3

The model's overestimating surface $O_3$ is studied in several publications. Young et al. (2013), using all the ACCMIP models, suggest that the high ozone bias in the Northern Hemisphere could indicate deficiencies with the ozone precursor emissions (see also for different models: Goldberg et al., 2016 and Travis et al., 2016). Moreover, in different experiences, for example, Lin et al. (2008) suggest that the model overestimation of ozone could also be due to an underestimation of ozone dry deposition velocity. In the same way, Ganzeveld et al. (2009) and Coleman et al. (2010) suggested that models are deficient in terms of dry deposition of gaseous species over oceans. Several other effects could be suggested such as a high sensitivity of models to meteorological fields (Hu et al., 2017) or a combination of excessive vertical mixing and net ozone production in the model boundary layer (see Travis et al., 2016). Then, we added a discussion in the new version of the paper (See L14-21 in Section 3.2):

> *"Our study is consistent with various modeling studies that have shown that models overestimate surface O3 observations at northern mid-latitudes. Young et al. (2013), using all the ACCMIP models, suggest that the high bias in the Northern Hemisphere could indicate deficiencies with the ozone precursor emissions (see also for different models: Goldberg et al. 2016 and Travis et al. 2016). Moreover, in different experiences, for example, Lin et al. (2008) suggest that the overestimation of models could also be due to an underestimation of ozone dry deposition velocity. In the same way, Ganzeveld et al. (2009) and Coleman et al. (2010) suggested that models are deficient in terms of dry deposition of gaseous species over oceans. Several other effects could be suggested such as a high sensitivity of models to meteorological fields (Hu et al. 2017) or a combination of excessive vertical mixing and net ozone production in the model boundary layer (see Travis et al. 2016)."*

**3) L235:** Please discuss the possible reasons for the best performance obtained by CMAM while comparing surface O3 over the Mediterranean Basin and ACCMIP simulations. The rational for referring to Parrish et al. (2014) in this context is not clear.

CMAM model shows better results for surface O3 in terms of error and bias compared to the other models but CMAM has a simplified chemical scheme (no NMVOCs). It is also important to keep in mind that a model simulation may appear close or far from the observations for the wrong reasons

(see for example Young et al., 2013) depending on the complexity and the non-linear processes implemented in the model. This kind of study can not explain the model performances. However, a sensitivity study may explain the differences between models, as well as their overestimation of surface ozone. As suggested by the reviewer, we added a possible reason for the best performance obtained by CMAM as:

> *"Note that CMAM model has a simplified chemical scheme (no NMVOCs). This may reduce uncertainties related to VOCs emissions."*

See L4-5 in Section 3.3. In addition, we removed the sentence that refer to Parrish et al. (2014).

**4) L298:** I don't see the point in discussing the ACCMIP inter-model variability of total VOC species since, as the authors mentioned, biogenic emissions are not specified and the VOC module differs from one model to another. The contribution of biogenic VOC (BVOC) emissions to regional ozone formation is significant. Over most of world's regions, the leading BVOC species, isoprene, reacts rapidly with hydroxyl radicals to form ozone (Pierce et al., 1998). The high ratio of BVOC to total VOC emission makes BVOC emissions a significant contributor to regional ozone concentrations. Estimations of BVOC emissions are subject to large uncertainty and are not well estimated by proxy concentrations (i.e. formaldehyde) observed from space. Hence, estimates of isoprene emissions by different biogenic emission models can vary more than a factor of two (Shim et al., 2005).

We agree with the reviewer that biogenic VOCs (BVOCs) contribute the most to the formation of ozone at regional scale. However, biogenic VOCs (BVOCs) are not specified in the framework of ACCMIP experiment but most of the models contain main BVOCs (in particular isoprene), except a few number of models (e. g. CMAM, HadGEM2). The total VOCs in the models are controlled by the BVOCs that are included in most of the models. To clarify this, we added this point in the new version of the paper.

We replace the following sentence:

> *"This is mainly due to two factors: (1) The VOC module is different from one model to another. In other words, some models have more VOC species than others, and especially isoprene is not included in a few models. (2) The second factor is that the biogenic emissions are not specified in the ACCMIP experiment."*

by (see L29-32 in section 4.2):

*"This is mainly due to two factors: (1) The VOC module is different from one model to another. In other words, some models have more VOC species than others, and especially isoprene is not included in a few models (CMAM and HadGEM2). (2) The second factor is that the biogenic emissions are not specified in the ACCMIP experiment (but are included in most of the models)."*

**5) L182-188:** The explanations for the high SD values over the Ligurian Sea and around Marseille are not clear. For example, do Vautard et al. (2007) give some evidences indicating that the stagnating conditions are the reason? What about the high SDs over the south-eastern parts of the Mediterranean Basin?

In fact, Vautard et al. (2007) only suggest that the over stagnation of air masses in ECMWF analyses in the Po valley may explain the very large values obtained for CHIMERE. That is why we cited this paper.
The standard deviation reaches 10 ppbv over the south-eastern part of the Mediterranean Basin, but this value is not really higher that the neighborhood values. In addition, in the new version of the paper, we added 2 models (MOCAGE and MIROC-CHEM), this relative weak maximum is not high-lighted anymore (see Figure 3b in the new version of the paper). Then, we think this is not necessary to discuss this point in the text. We prefer to focus on the predominant maximum in the Ligurian sea.

**6) L160:** I wonder, whether is not it obvious that a comparison of models to surface O3 averages for such long periods (8 to 16 years and for 1990-2010) is expected to give relative elevated correlations?

It is true that the correlation coefficients between modeled and observed surface $O_3$ are relatively high for the annual cycle. Young et al. (2013) obtained scores of the same order at the global scale, using almost the same set of model outputs. They found that the correlation coefficients for annual cycle varying between 0.89 and 0.97 at 700 hPa (Young et al., 2013, Figure 4). However, there is still a disparity between models in terms of correlation coefficient that show that is not so obvious to get such an elevated correlation (see Taylor diagram in Figure 2b).

**Minor comments**

**1) L153-154:** The maximum observed concentration recorded in July as compared to the ACCMP models August average for 1990-2010 contradicts the statement made by the authors in line 154.

We added the following sentence to clarify this point (see L8-9 in section 3.1): *"It should also be noted that the GEOSCCM, GISS-E2R, EMAC, HadGEM2 and LMDZ-OR-INCA models show a maximum of surface ozone concentrations in August, contrary to the observations that show a maximum in July"*.

**2) L175:** Please replace "Northern Africa" by south-east Mediterranean Basin.

We agree, the sentence was confusing. We rewrote the sentence as follows (see L25-28 in section 3.2): *"The general features, with higher O3 concentrations over the MB and the Middle East region, are observed, exceeding an average of 60 ppbv in the center of the MB. Over the continental Europe and Northern Africa, the surface O3 concentrations are smaller (≈40 ppbv) than over the MB"*.

**3) L377:** Please supply further explanations for the term "highly dynamical".

We replace the following sentence:

*"which is a highly dynamical region (Lelieveld et al., 2002, 2010)."*

by (see L5-7 in section 4.4):

*"which is directly under the descending branch of the Hadley circulation, particularly in summer, driven by deep convection in the Inter-Tropical Convergence Zone (Lelieveld et al., 2002)."*

**4) L389:** The sentence: "This shows that a positive correlation between changes in CH4 and surface O3 can exist" is not clear, please elaborate.

We agree with the reviewer, the sentence was confusing and we replace the following sentence :

[revised manuscript text omitted]

**Global scale**

| MnB | MAGE | RMSE | NMBF | NMAEF | |
|---|---|---|---|---|---|
| 15 | 15 | 17 | 0.49 | 0.5 | GFDL–AM3 |
| 8 | 10 | 13 | 0.26 | 0.34 | CESM–CAM–superfast |
| 0.39 | 6.2 | 8.2 | 0.013 | 0.21 | CMAM |
| 19 | 20 | 22 | 0.64 | 0.65 | GISS–E2–R |
| 5.4 | 7.5 | 10 | 0.18 | 0.25 | EMAC |
| 9.7 | 11 | 13 | 0.32 | 0.35 | GEOSCCM |
| 7.4 | 11 | 14 | 0.25 | 0.37 | LMDz–OR–INCA |
| 8.2 | 9.4 | 12 | 0.27 | 0.31 | NCAR–CAM3.5 |
| 6 | 8.8 | 11 | 0.2 | 0.29 | MIROC–CHEM |
| 9.8 | 11 | 14 | 0.33 | 0.37 | UM–CAM |
| −3.4 | 8.3 | 10 | −0.13 | 0.31 | HadGEM2 |
| 4.6 | 8 | 10 | 0.15 | 0.27 | STOC–HadAM3 |
| 9.6 | 12 | 17 | 0.32 | 0.42 | MOCAGE |

**Regional scale**

[revised manuscript text omitted]

---

## Author Comment (AC2) · 4 Jan 2018

Responses to reviewer 2

Future changes in surface ozone over the Mediterranean basin in the framework of the Chemistry-Aerosol Mediterranean Experiment (ChArMEx)

R2) This is a paper of interest to the CHARMEX community, providing a reasonably standard analysis of a series of models and their performance over the Mediterranean basin for various scenarios (RCP; time slices). However, I find that the paper describes many model results, but does not attempt sufficiently to explain their performance (e.g., why they differ from observations, why there are outliers).

Furthermore, I see little evidence explaining the credibility of the model results for the future scenarios.

For this reason, in my view the paper is not ready for publication in ACP. The authors should try to address these points, perhaps engaging in intelligent speculation if needed. The authors should also address the specific comments below.

We thank the reviewer for his/her valuable comments. In this new version of manuscript, we present all available data (including outliers) in the reference period (1990-2010). In particular, our statistics include the results from MOCAGE and MIROC-CHEM, increasing the number of models from 11 to 13 models. The figures and the values are updated accordingly in this new version of paper. Below, we answer point to point to the specific comments and to the general comment below:

The aim of this paper is to study the future evolution of surface ozone over the Mediterranean Basin with a focus on summertime over the time period 2000-2100, using the outputs of the various AC-CMIP models. Explaining the model performance is not the purpose of the paper, we only used the model outputs to study the future changes in surface ozone. The present-day surface ozone period (called 2000 period representing the years between 1990 and 2010) from ACCMIP models was compared to observations issued from Sofen et al. (2016) and the evaluation was carried out in order to evaluate uncertainties related to model simulations during the contemporary period (2000-2010). The model evaluation is an important step to describe model strengths and weaknesses and to ensure their ability to reasonably reproduce the future evolution of simulated surface ozone. It is true that the models are distinct and each model has its own characteristics (in particular, transport and chemical schemes), but the performance of a model (as defined by the reviewer) cannot be simply explained by such a study. We only are users of the different ACCMIP model outputs. However, a sensitivity study with regards to the different model parameters (chemical schemes,

transport, emissions…) may explain the differences between models, but this is out of scope of our paper.

Concerning the credibility of the ACCMIP models for the future scenarios, ACCMIP models are forced by the latest generation of scenarios (RCPs scenarios). For most variables, theses RCPs cover a wide range of the existing literature. The RCPs are an important development in climate research and provide a potential foundation for further research and assessment, including emissions mitigation and impact analysis (Van Vuuren et al., 2011). In this paper, we analyse simulations performed from a set of models under the four existing RCPs to be able to investigate the future changes in surface O3 under a wide range of future projections. We can point out at least 3 relevant points:

**1)** The ACCMIP models have been studied in numerous publications (e.g. Young et al., 2013 and Silva et al., 2016) to investigate the evolution and distribution of tropospheric ozone for a range of RCPs showing confidence in the results. Young et al. (2013) show that the relative changes for the global tropospheric ozone burden in 2100 decrease for the RCP2.6, RCP4.5 and RCP6.0 and increase for the RCP8.5. Silva et al. (2016) show that the change in future ozone concentrations relative to 2000 is associated with excess global premature mortality in some scenarios/periods, particularly in RCP8.5 in 2100 (316 thousand deaths per year).

**2)** The statistical tests made in this paper for most of the parameters show trends which are statistically significant (see Figure R2.1 and R2.2).

**3)** The surface ozone evolution is consistent for most of the ACCMIP models over the Mediterranean Basin. In the new version of the paper, all the ACCMIP models are presented (over the Mediterranean Basin), even the outliers.

<h1 align="center">Specific comments</h1>

**P. 3: L. 22:** With multiple citations stick to either an alphabetical scheme or a chronological scheme, but do not mix them.  Done

**P. 6 L. 84:** uses -> use.  Done

**P. 9 L. 158:** Introduce acronym for ppbv. Make sure all acronyms are introduced, both in the abstract and in the main manuscript. Done.

**P. 10 L. 171:** Why do these models represent best the annual cycle of surface O3?

This sentence is confusing, and is removed in the new version of the paper.

**P. 12 L. 214:** Why do these models show less variability than the observations?

The variability of the models is less because of the emissions inventory does not change year to year over the 2000 period. Then, to compare the model variability with the variability of observations does not show any concluding results. For this reason, we removed this section which does not bring any added value to our paper.

**P. 13 L. 239-242:** What is the reason for this behaviour in the models?

Several studies have highlighted a high bias in the Northern Hemisphere and a low bias in the Southern Hemisphere. This bias appears systematically in most models, suggesting that a deficiency in emissions can occur as a part of this bias in the outputs of the models (e.g. Young et al., 2013; Travis et al., 2016).

**P. 14 L. 256:** Is this trend significant?

We calculated the student T test for the 95% confidence level and we show that trends in temperature for all RCPs between 2000 and 2100 are statistically significant. The p-value is the probability to have the same sample mean for 2000 and 2100. When the p-value is less than 5%, the means are statistically different between the samples. In figure below, the p-value indicates the trend significance. For example, for RCP2.6 the p-value is 0.029 less than 0.05 which means that trend between 2000 and 2100 is statistically significant (see Figure R2.1).

[Figure]

*Figure R2.1* Box-whisker plots of the annual average of temperature in kelvin (K) since 2000, calculated over the Mediterranean Basin domain for the JJA period and for the RCP2.6 (yellow), RCP4.5 (green), RCP6.0 (blue) and RCP8.5 (red). The median is indicated by the thick horizontal black line, the multi-model mean by a filled diamond, the (25-75%) range by the colored box and minimum/maximum excluding outliers by whisker. Each filled circle represents a single model. The p-value of the student T test at the 95% confidence level is indicated for each RCP trend between 2000 and 2100 at the top of the figure. This means that if the p-value between two groups is less than 0.05, the trend is significant.

**L. 267:** probably -> likely.  Done

**P. 15 L. 279:** asset -> assets.  Done

**P. 16 L. 306:** Why are these models outliers?

The overestimation of CO emissions by HadGEM2 is mainly due to the fact that this model includes extra CO emissions as a substitute replacement for missing non-methane volatile organic compounds (NMVOCs).

**P. 17 L. 314:** You mean the simulated emissions, correct?

Emissions are not simulated, but described for the model simulations. Ozone precursor emissions from anthropogenic and biomass burning sources were taken from those compiled by Lamarque et al. (2010) for the Hist simulations, whereas emissions for the RCP simulations were developed by four modeling teams. Each of them applied a set of algorithms to ensure consistency with the 2000 emission inventory (Lamarque et al., 2013).

**P. 19 L. 364-370:** How credible are these model results?

These are conclusions from the analysis done in section 4.3. Once again, we only used model outputs from an international comparison exercise which was made using a specific protocol (see Lamarque et al., 2013). However, in this partial conclusion, there are two results: (i) the future changes in surface ozone (in terms of mean) over the Mediterranean Basin; (ii) the spatial future changes in surface ozone over the Mediterranean region (see Figure 9 in the new version of the paper).

Concerning (i), the surface ozone trends are statistically significant (see figure below). The p-value using the Student T test for 95% confidence level is indicated for each RCP trend between 2000 and 2100 and presents values less than 0.05 for RCP2.6, RCP4.5 and RCP6.0. For RCP8.5, the p-value is greater than 0.05 which means that the multimodel mean is constant between 2000 and 2100.

Concerning (ii), we reported that the statistical significance of trends was already evaluated in P17-L331-332 (P10–L31-32 in the new version of the paper) and see Figure R2.2.

[Figure]

*Figure R2.2:* legend Box-whisker plots of the annual average of surface ozone in ppbv since 2000, calculated over the Mediterranean Basin domain for the JJA period and for the RCP2.6 (yellow), RCP4.5 (green), RCP6.0 (blue) and RCP8.5 (red). The median is indicated by the thick horizontal black line, the multi-model mean by a filled diamond, the (25-75%) range by the colored box and minimum/maximum excluding outliers by whisker. Each filled circle represents a single model. The p-value of the student T test at the 95% confidence level is indicated for each RCP trend between 2000 and 2100 at the top of the figure. This means that if the p-value between two groups is less than 0.05, the trend is significant.

**P. 23 Conclusions:** The authors should consider splitting this into a discussion section and a short conclusions section.

As the reviewer, we think the discussion part is important. Nevertheless, our approach consists in discussing the results in section 4 and in summarizing the main results in the conclusions. For these reasons, we do not think that splitting our conclusion into a discussion and a short conclusion is adequate. However, in the introduction of section 4, we clarify this point by introducing the discussion.

**Table 1:** The paper mentions 11 models, but the table suggests that less are used. Could the authors clarify this.

Thank you for this remark, we corrected the table and we added all missing data into the new version of the paper. Note we also added the outliers that likely show the limits for the different trends.

**Fig. 4 caption:** I suggest you indicate what the endpoints of the colour scale mean. Same for similar figures. Done

**Fig, 6 caption:** Indicate what the coloured ellipses mean. The figure has been removed (see answer 5 of specific comment)

**Fig. 7 caption:** "...The metrics used..."  Done

**Fig. 8 caption:** Identify the colour scheme in the caption. Same for Figs. 12, 13. Done
- Fig 8 caption (Figure 6 in the new version of the paper): "for the four Representative Concentration Pathways" was replaced by "for the RCP2.6 (yellow), RCP4.5 (green), RCP6.0 (blue) and RCP8.5 (red)".
- Fig.12 caption (Figure 10 in the new version of the paper): "for the four Representative Concentration Pathways" was replaced by "for the RCP2.6 (yellow), RCP4.5 (green), RCP6.0 (blue) and RCP8.5 (red)".
- Fig 13 caption (Figure 11 in the new version of the paper): "for the RCP2.6, RCP4.5 and RCP6.0" was replaced by "
[revised manuscript text omitted]

**Global scale**

| MnB | MAGE | RMSE | NMBF | NMAEF | |
|---|---|---|---|---|---|
| 15 | 15 | 17 | 0.49 | 0.5 | GFDL–AM3 |
| 8 | 10 | 13 | 0.26 | 0.34 | CESM–CAM–superfast |
| 0.39 | 6.2 | 8.2 | 0.013 | 0.21 | CMAM |
| 19 | 20 | 22 | 0.64 | 0.65 | GISS–E2–R |
| 5.4 | 7.5 | 10 | 0.18 | 0.25 | EMAC |
| 9.7 | 11 | 13 | 0.32 | 0.35 | GEOSCCM |
| 7.4 | 11 | 14 | 0.25 | 0.37 | LMDz–OR–INCA |
| 8.2 | 9.4 | 12 | 0.27 | 0.31 | NCAR–CAM3.5 |
| 6 | 8.8 | 11 | 0.2 | 0.29 | MIROC–CHEM |
| 9.8 | 11 | 14 | 0.33 | 0.37 | UM–CAM |
| −3.4 | 8.3 | 10 | −0.13 | 0.31 | HadGEM2 |
| 4.6 | 8 | 10 | 0.15 | 0.27 | STOC–HadAM3 |
| 9.6 | 12 | 17 | 0.32 | 0.42 | MOCAGE |

**Regional scale**

[revised manuscript text omitted]

---

## Referee Report (RR1)

**Review of Jaidan et al. "Future changes in surface ozone over the Mediterranean Basin in the framework of ChArMEx"**
Re-submission to ACP, 2018

Note: My review is of the revised version of this manuscript, not having seen the original submission.

This manuscript presents a study of the present and projected future of surface ozone over the Mediterranean Basin, as simulated by a range of global chemistry models that took part in the ACCMIP experiment. It is no doubt a useful contribution to ChArMEx and is broadly interesting inasmuch as surface ozone projections from global models are being used in impact studies and international (climate) reports.

Overall, my opinion is that the manuscript needs a further iteration of revisions. Below, I have made some comments on the authors' response to reviewers (Section A), followed by specific comments on the revised manuscript (Section B), and technical corrections after that (Section C).

**A. Comments on the authors' responses to reviewers:**
1. I agree with the authors' comments that a detailed explanation of the drivers of model biases/differences is not feasible. While this appears unsatisfactory to some, to do this properly in models with 1000s of parameters would require a substantial (albeit necessary) research effort, organised across multiple modelling centres. It's not just the emissions, deposition and chemistry scheme, but also physics parameters in the underlying GCM, including biases in (e.g.) the timing and location of winds, clouds, temperatures, rainfall etc.

There are efforts underway to better understand the interaction of all the biases, but we must recognise that we are dealing with phenomena that emerge from a complex interaction of multiple processes and knowing that models are "right for the right reason" will be a fraught question.

2. There seems to be some confusion about assessing statistical significance, T-test and p-values, at least as written (e.g., bottom of P9 of the response). One does not "calculate the student T test for the 95% confidence level"; rather the Student's t test gives the t-statistic, which - for a given number of degrees of freedom - can then be used to give a p-value (e.g., by using statistical software). See also my specific comments above, related to the graphs.

Additionally, I would not call it a "trend" between 2000 and 2100 as it is really a different between two time slices.

**B. Specific Comments on the manuscript**

P2, L33: Some of the ACCMIP models were not chemistry-climate models (e.g., CICERO-CTM2 is a CTM, and MOCAGE and STOCHEM are basically run as CTMs - see the Young et al. ACCMIP paper).

P3, L8 (and for general consideration): There no mention of the hourly ozone output as part of ACCMIP, which might add some further depth (or at least context) to the analysis. It would at least be good to mention the analysis and conclusions of Schnell et al. (2015, ACP, doi: 10.5194/acp-15-10581-2015), who looked at this in the context of AQ in Europe and N America.

P3, L32: See Iglesias-Suarez et al. (2016, ACP, doi: 10.5194/acp-16-343-2016) for a description and evaluation of stratospheric ozone in the ACCMIP models.

P6, L12: How can the mean "simulate appropriately", yet have "a consistent positive bias"?

P7, L4: What types of models did Vautard et al. evaluate? Is their conclusion likely to be valid for ACCMIP?

P9, L4-5: Sentence starts saying "Several studies" and then only references one at the end.

P10, L2-5: The authors mention later, but here it would be good to note that there is considerable variability in the complexity of the VOC scheme (and total emissions of reactive C) between the ACCMIP models. See figure of the emissions in Young et al. (2013).

P10, L30: "We use the Student t test for the 95% confidence interval...". Either the grammar here is wrong, or there's perhaps a misunderstanding about the t test - see comment #2 in Section B.

P12, Section 4.4: I'm afraid I found this section very hard going to understand, and I wonder if it could be re-worded to be clear about what trends are from precursors and what are from climate? (See also my comment about paragraphs below).

For the impact of climate, why did the authors not analyse the subset of ACCMIP models that completed sensitivity studies with fixed emissions? See Stevenson et al. (2013; ACP, doi: 10.5194/acp-13-3063-2013).

P14, Conclusions: This section appears to be rather a laundry list of individual results, with no synthesis and little in the way of outlook. What should people doing impact studies take away from this analysis, for instance?

Figure 2 (and related discussion): Is the seasonal cycle consistent for all the grid squares in this evaluation? Is there any interannual variability in the observations that should be used on the error bars? (The models were not simulating the meteorology for the year 2000, so the comparison needs to be applied fairly, somehow).

Figure 3: Please try and avoid the rainbow colour scale (e.g., see http://bit.ly/2rN9RjM; applies to other figures too). Also, what is gained from having so many individual levels? Can anyone tell the difference between 23 shades of blue? Finally, please state whether the standard deviation is the intermodel spread, or temporal. (I guess the former, but it's ambiguous.)

Figure 5: A colour bar for the table might be useful, even if it is just qualitative. …Is it based on ranking?

Figure 6: Caption starts by saying annual average, when it is a JJA average. …Also, if you are showing absolute numbers (are you sure you want to do that?), then it would be good to show comparison numbers from (e.g.) a reanalysis product. Climate models are biased for the global mean, so I am sure that they will be so for a smaller region.

Figure 7: Please put (a), (b) etc before the species to which it refers.

Figure 9: This figure is very small, and (similar to my comment on Figure 3), I think the colour bar colours and levels needs revisiting. Furthermore, have the authors considered the "field significance" in their indication of significant (or not) differences? See Wilks (2016, BAMS, doi: 10.1175/BAMS-D-15-00267.1).

Figure 11 and 12: Is a box-whisker plot appropriate for 5 models?

**C. Technical corrections to the manuscript**
1. There are an awful lot of very long paragraphs. Please split up the text for ease of reading. E.g., P1,L10: new paragraph at "Tropospheric..." (and combine with next shorter paragraph; P1 L31: new paragraph at "A number..." etc.

2. A proof read would help. E.g., P2, L18:  "...usually observed in summer period" -> "...usually observed in THE summer period"; Pp, L13: "experience" -> "experiment"

3. Throughout (for consideration): Why write "O3" instead of "ozone"? We say the latter; we don't say "o-3". This helps readability in my view.

---

## Author Response (AR2)

**Responses to reviewer 1**

**Future changes in surface ozone over the Mediterranean basin in the framework of the Chemistry-Aerosol Mediterranean Experiment (ChArMEx)**

Note: My review is of the revised version of this manuscript, not having seen the original submission.

This manuscript presents a study of the present and projected future of surface ozone over the Mediterranean Basin, as simulated by a range of global chemistry models that took part in the ACCMIP experiment. It is no doubt a useful contribution to ChArMEx and is broadly interesting inasmuch as surface ozone projections from global models are being used in impact studies and international (climate) reports.

Overall, my opinion is that the manuscript needs a further iteration of revisions. Below, I have made some comments on the authors' response to reviewers (Section A), followed by specific comments on the revised manuscript (Section B), and technical corrections after that (Section C).

We thank the reviewer for his/her valuable comments. We answer point to point to the comments in blue:

**A. Comments on the authors' responses to reviewers:**

**1**. I agree with the authors' comments that a detailed explanation of the drivers of model biases/differences is not feasible. While this appears unsatisfactory to some, to do this properly in models with 1000s of parameters would require a substantial (albeit necessary) research effort, organised across multiple modelling centres. It's not just the emissions, deposition and chemistry scheme, but also physics parameters in the underlying GCM, including biases in (e.g.) the timing and location of winds, clouds, temperatures, rainfall etc.

There are efforts underway to better understand the interaction of all the biases, but we must recognise that we are dealing with phenomena that emerge from a complex interaction of multiple processes and knowing that models are "right for the right reason" will be a fraught question.

**2**. There seems to be some confusion about assessing statistical significance, T-test and p- values, at least as written (e.g., bottom of P9 of the response). One does not "calculate the student T test for the 95% confidence level"; rather the Student's t test gives the t-statistic, which - for a given number of degrees of freedom - can then be used to give a p-value (e.g., by using statistical software). See also my specific comments above, related to the graphs.

Additionally, I would not call it a "trend" between 2000 and 2100 as it is really a different between two time slices.

We agree with the reviewer that the student T test gives us a p-value from which we can determine if the test is significant or not according to a chosen confidence level.

In addition, as suggested by the reviewer, we changed the term "trend" as necessary throughout the paper.

**B. Specific Comments on the manuscript**

P2, L33: Some of the ACCMIP models were not chemistry-climate models (e.g., CICERO- CTM2 is a CTM, and MOCAGE and STOCHEM are basically run as CTMs - see the Young et al. ACCMIP paper).

We replace the following sentence:

"The assessment of the future changes in annual tropospheric O3 at global scale has been done by Young et al. (2013) using a set of chemistry-climate models"

by (see P3.L4 in section 1):

"The assessment of the future changes in annual tropospheric ozone at the global scale has been done by Young et al. (2013) using a set of chemistry models."

To clarify this, we added this point in the new version of the paper (See P3.L27-30 in Section 2.1):

"Most of the models we used are chemistry climate models (CCMs) except three models:  MOCAGE which is a chemical transport model (CTM), using off-line meteorological fields from an appropriate simulation of a climate model; STOC-HadAM3 and UM-CAM, referred as chemistry-general circulation models (CGCMs), which produce their own meteorology without any interaction with climate."

P3, L8 (and for general consideration): There no mention of the hourly ozone output as part of ACCMIP, which might add some further depth (or at least context) to the analysis. It would at least be good to mention the analysis and conclusions of Schnell et al. (2015, ACP, doi: 10.5194/acp-15-10581-2015), who looked at this in the context of AQ in Europe and N America.

We agree with the reviewer, we updated the new version of the paper by

including the following sentence (see L31+ in section 3.2):

"Schnell et al. (2015) evaluated a set of ACCMIP models against hourly surface ozone from 4217 ground based stations in North America and Europe. They found that models are generally biased high during all hours of the day and in all regions. Moreover, they also found that most models well simulate the shape of regional summertime diurnal and annual cycles. They concluded that the skill of the ACCMIP models provides confidence in their projections of future surface ozone."

In addition we added the reference Schnell et al (2015) as suggested by the review

P3, L32: See Iglesias-Suarez et al. (2016, ACP, doi: 10.5194/acp-16-343-2016) for a description and evaluation of stratospheric ozone in the ACCMIP models.

We updated the new version of the paper by including the following sentence (see P4.L6-L9 in section 2.1):

"Iglesias-Suarez et al. (2015) evaluated the stratospheric ozone and associated climate impacts using the ACCMIP simulations in the recent past (1980–2000). They showed that ACCMIP multi-model mean total column ozone trends compare favorably against observations. They also demonstrated how changes in stratospheric ozone are intrinsically linked to climate changes".

In addition, we added the reference Iglesias-Suarez et al. (2016) as suggested by the review.

P6, L12: How can the mean "simulate appropriately", yet have "a consistent positive bias"?

The sentence is misleading and we changed it by (see P6.L19-21 in section 3.1): " The behavior of the annual cycle of surface ozone from ACCMIP models averaged over the period 1990-2010 over the Mediterranean basin is quite similar to the one observed. The bias between the ACCMIP and the observed annual cycle is positive with values between 6.10 and 12.47 ppbv."

P7, L4: What types of models did Vautard et al. evaluate? Is their conclusion likely to be valid for ACCMIP?

Vautard evaluated six different chemistry transport models over a full year (1999). Three models are used both at large-scale (typically 50 km) and small-scale resolution (5 km).

The results from Vautard indicate the importance of the meteorological forcings that induce a difference between the model results in the region of Po-Valley. We just used this result to provide a possible reason for the disagreement of the ACCMIP models in this specifically sensitive region.

P9, L4-5: Sentence starts saying "Several studies" and then only references one at the end.

We replace the following sentence:

"Several studies have shown that humidity is the most important meteorological factor affecting OH and CH4 lifetimes (Spivakovsky et al., 2000), which are involved in the chemical production of O3."

by (see P9.L18-19 in section 4.1):

"Spivakovsky et al. (2000) showed that humidity is the most important meteorological factor affecting the lifetimes of OH and CH4 which are involved in the chemical production of ozone."

P10, L2-5: The authors mention later, but here it would be good to note that there is considerable variability in the complexity of the VOC scheme (and total emissions of reactive C) between the ACCMIP models. See figure of

the emissions in Young et al. (2013).

To clarify this, we added this sentence in the new version of the paper (See P10.L15-16 in Section 4.2):

"Note that there is considerable variability in the complexity of the chemical schemes, in particular for the VOC schemes between the ACCMIP models."

P10, L30: "We use the Student t test for the 95% confidence interval...". Either the grammar here is wrong, or there's perhaps a misunderstanding about the t test - see comment #2 in Section B.

We corrected the sentence by "We use a Student's T-test with a 95% confidence interval..." see also our answer for the comment #2 in Section B.

P12, Section 4.4: I'm afraid I found this section very hard going to understand, and I wonder if it could be re-worded to be clear about what trends are from precursors and what are from climate? (See also my comment about paragraphs below).

For the impact of climate, why did the authors not analyse the subset of ACCMIP models that completed sensitivity studies with fixed emissions? See Stevenson et al. (2013; ACP, doi: 10.5194/acp-13-3063-2013).

We reworded parts of the section 4.4 as suggested by the reviewer. Our purpose was to focus on the effects of ozone precursors in the context of climate change. We also changed the title to clarify this point.

We agree with the reviewer, it is interesting to study the impact of the climate change by using these sensitivity simulations. However, our goal was to compare the results between the different scenarios by keeping the largest number of models (and only 6 models have provided outputs for the sensitivity study following the RCP8.5 scenario).

P14, Conclusions: This section appears to be rather a laundry list of individual results, with no synthesis and little in the way of outlook. What should people doing impact studies take away from this analysis, for instance?

We rearrange the conclusion to highlight the message of our paper. We added this paragraph in the end of the conclusion:

"The surface ozone decrease over the MB for the scenarios RCP2.6, RCP4.5 and RCP6.0 is much more pronounced than the relative changes of the global tropospheric ozone burden. This reflects the fact that the surface ozone over the MB is more controlled by reductions of its precursor emissions, water vapor represented by the increase in the specific humidity and the NOx-limited regime over the MB. In this region, for the RCP8.5 scenario, we showed how the future climate change and in particular the increase in methane concentrations can offset the benefit of the reduction in emissions of ozone precursors. Future modeling studies should quantify the sensitivity of the future surface ozone to climate change and methane concentrations changes over the MB"

Figure 2 (and related discussion): Is the seasonal cycle consistent for all the grid squares in this evaluation? Is there any interannual variability in the observations that should be used on the error bars? (The models were not simulating the meteorology for the year 2000, so the comparison needs to be applied fairly, somehow).

1) This is a difficult point to answer because we combined spatial and inter-model mean. We have preferred to focus our study on the ozone variability between the ACCMIP models rather than on the ozone variability in the very small domain only covering the MB.

2) We added in Figure 2, the standard deviation of the observations that show the variability of the observations.

Figure 3: Please try and avoid the rainbow colour scale (e.g., see http://bit.ly/2rN9RjM; applies to other figures too). Also, what is gained from having so many individual levels? Can anyone tell the difference between 23 shades of blue? Finally, please state whether the standard deviation is the intermodel spread, or temporal. (I guess the former, but it's ambiguous.)

We changed the rainbow colorbar by another colorbar taking into account the small number of the different levels as suggested by the reviewer.

We changed the term "standard deviation" by "inter-model standard deviation" to clarify the sentence.

Figure 5: A colour bar for the table might be useful, even if it is just qualitative. ...Is it based on ranking?

Yes it is based on ranking; the colorbar goes from close to the observation to far from the observation for each metric. We added a qualitative colobar in Figure 5.

Figure 6: Caption starts by saying annual average, when it is a JJA average. ...Also, if you are showing absolute numbers (are you sure you want to do that?), then it would be good to show comparison numbers from (e.g.) a reanalysis product. Climate models are biased for the global mean, so I am sure that they will be so for a smaller region.

We corrected the sentence by changing "annual average" to "summer (JJA) average".

We are not quite sure to understand the point of view from the reviewer but we only could use reanalysis for the contemporary period. From this figure, we can compare the different box plots to the reference (REF). In addition, our study is not focused on the meteorological parameters. We plotted the absolute values to have an idea of the amplitude of ACCMIP models for each parameter. However, we are interested in the parameter difference for two periods (2030 and 2100) in the future to put into evidence the link

between the meteorological parameters evolution and the one of ozone for ACCMIP models.

Figure 7: Please put (a), (b) etc before the species to which it refers.

Corrected

Figure 9: This figure is very small, and (similar to my comment on Figure 3), I think the colour bar colours and levels needs revisiting. Furthermore, have the authors considered the "field significance" in their indication of significant (or not) differences? See Wilks (2016, BAMS, doi: 10.1175/BAMS-D-15-00267.1).

We changed the colorbar. We do not consider the field of significance, but we use local tests to have an idea on the statistical significance of surface ozone changes, as we mentioned on page P11.L13-14.

Figure 11 and 12: Is a box-whisker plot appropriate for 5 models?

We agree with the reviewer that it is more appropriate to use a box plot when the number of models is relatively high. However, we find that we have additional information such as the mean and median. As well as the figure reading is easier to understand with the colors.

**C. Technical corrections to the manuscript**

1. There are an awful lot of very long paragraphs. Please split up the text for ease of reading. E.g., P1,L10: new paragraph at "Tropospheric..." (and combine with next shorter paragraph; P1 L31: new paragraph at "A number..." etc.

Done

2. A proof read would help. E.g., P2, L18: "...usually observed in summer period" -> "...usually observed in THE summer period"; Pp, L13: "experience" -> "experiment"

We reread the paper and corrected the paper as much as possible. Concerning this example, we changed "usually observed in summer period" by "usually observed in the summer period" and "experience" by "experiment".

3. Throughout (for consideration): Why write "O3" instead of "ozone"? We say the latter; we don't say "o-3". This helps readability in my view.

We changed O3 by ozone throughout the paper.

**Future changes in surface ozone over the Mediterranean basin in the framework of the Chemistry-Aerosol Mediterranean Experiment (ChArMEx)**

The authors start to address my previous comments. However, like the other reviewer, I still think that this paper needs further work before it is suitable for publication in ACP. One thing the authors should do is improve the readability of the text by doing the following. (i) Reduce the size of the paragraphs (which are very long) – perhaps by splitting them into smaller units. (ii) Look at the english – in particular, there are quite a few typos. (iii) Rewrite the conclusions, so that they are less of a summary of the results, and more of a reflection of the strengths and weaknesses of the models, in this case for studying climate change in the Mediterranean Basin.

The authors should also address the specific comments below.

We thank the reviewer for his/her valuable comments. We answer point to point to the specific comments in blue:

**Specific comments**

P. 1 L. 8-10: Maybe I am missing something, but the text suggests the model ensemble mean simulates well the annual cycle of surface ozone, but that a majority of the models overestimate the surface ozone during the period 2000-2010 and for summer. Is this behaviour consistent? Perhaps you need a clarification here and elsewhere in the paper.

We mean that the models simulate well the behavior of the annual surface ozone cycle. We have clarified this in the updated paper (see P6.L19-21 in section 3.1).

L. 15: Where do these increases in CH4 come from?

$CH_4$ emissions evolution was specified for each of the RCPs. For the RCP8.5, $CH_4$ emissions will increase between 2000 and 2100 (Van vuuren et al., 2011). In addition, increases in life-stock population, rice production, and enteric fermentation processes drive emissions of methane (Riahi et al., 2011).

P. 2 L. 9: Perhaps include references for these sinks.

As suggested by the reviewer, we added the following reference: Jacob (2000)

P. 3 L. 3: I understand from the list of models that not all are climate-chemistry models. Please clarify. The other referee also made this comment.

We replace the following sentence:

"The assessment of the future changes in annual tropospheric O3 at global scale has been done by Young et al. (2013) using a set of chemistry-climate models"

by (see P3.L4 in section 1):

"The assessment of the future changes in annual tropospheric ozone at the global scale has been done by Young et al. (2013) using a set of chemistry models."

To clarify this, we added this sentence in the new version of the paper (See P3.L27-30 in Section 2.1):

"Most of the models we used are chemistry climate models (CCMs) except three models: MOCAGE which is a chemical transport model (CTM), using off-line meteorological fields from an appropriate simulation of a climate model; STOC-HadAM3 and UM-CAM, referred as chemistry-general circulation models (CGCMs), which produce their own meteorology without any interaction with climate."

P. 5 L. 28: A style point: consider not starting a sentence with an acronym.

Done

P. 8 L. 4: Do you need "obviously"? Omit needless words. Same for "in fact" on P. 9, L. 29.

We corrected the different sentences.

P. 9 L. 6: You mention several studies, but only provide one reference.

We replace the following sentence:

"Several studies have shown that humidity is the most important meteorological factor affecting OH and CH4 lifetimes (Spivakovsky et al., 2000), which are involved in the chemical production of O3."

by (see P9.L19-20 in section 4.1):

"Spivakovsky et al. (2000) showed that humidity is the most important meteorological factor affecting the lifetimes of OH and CH4 which are involved in the chemical production of ozone."

L. 12: I am not sure I understand your statement "according to the radiative forcing". You mean there is a direct relationship between the temperature increases and the different radiative forcings? Please clarify.

We agree with the reviewer that the relationship between the temperature increase and the different radiative forcings are not direct. We clarified this by

adding this sentence: "Even it is not a direct relationship, we note that the temperature rises with increased radiative forcing "

P.10 L. 15-19: Quantify the statements made. I suggest you do not use words like "drastically". Please avoid hyperbolic language. Do this elsewhere in the paper.

Done

P. 11 L. 30+: Quantify the statements made.

Done

P. 12 L. 29: I suggest you use "complicated" rather than "complex".

Done

P. 13 L. 6-10: Quantify these statements.

Done

L. 14: Perhaps remind the reader of these scenarios and periods.

Done

L. 24-25: What do you mean by increasing chemical terms? Which chemical terms?

What we called the chemical terms are the chemical production (P) the chemical loss (L) and the chemical budget (P-L). We have specified this in the updated version.

P. 14 Section 5: This is too long and it is difficult to see what inferences one can make about the capability of the models to simulate climate change over the Mediterranean basin. There is just one line at the end of this section. The authors should address this.

We have clarified Section 5

Fig. 12: This is not quite the same as Fig. 11. Please reword. ??

We reworded the figure 11 caption by:

"Future relative change in surface ozone budget over the MB domain for JJA period and for the RCP8.5: (a) chemical production (P), (b) chemical loss (L) and (c) chemical budget (P-L) of surface ozone, (d) dry deposition of ozone (D)

The median is indicated by the thick horizontal black line, the multi models mean by a filled diamond, the (25-75%) range by the colored box and minimum/maximum excluding outliers by whisker. Each point represents a single model. The dashed horizontal line represents the mean for the REF period (2000) considered as a reference."

Table 3: Identify which are chemistry-climate models and which are chemistry-transport models.

In Table 3, we have added a column named "type" that shows the category of each model. In addition, we also have added this point in the new version of the paper (See P3.L27-30 in Section 2.1):

"Most of the models we used are chemistry climate models (CCMs) except three models: MOCAGE which is a chemical transport model (CTM), using off-line meteorological fields from an appropriate simulation of a climate model; STOC-HadAM3 and UM-CAM, referred as chemistry-general circulation models (CGCMs), which produce their own meteorology without any interaction with climate."

---

## Author Response (AR3)

Responses to reviewer 1

We thank the reviewer for his/her valuable comments. We answer point to point to the comments in blue:

The authors have untaken several efforts to address the previous review comments and I think the paper is improved. Below I have provided some specific comments on the revised manuscript as well as a comment on their response, and some general comments.

COMMENT ON AUTHORS' RESPONSE 1.

The authors were not sure about my comment on figure 6 in my previous review, repeated here: "If you are showing absolute numbers (are you sure you want to do that?), then it would be good to show comparison numbers from (e.g.) a reanalysis product. Climate models are biased for the global mean, so I am sure that they will be so for a smaller region." What I meant was that the physical climate world normally deals in anomalies, or changes in parameters from a given period. One reason to do this is that the models are biased for the absolute value, but they might get a trend that agrees with the observations. The authors are showing absolute values for temperature, humidity and rainfall, whereas differences to (say) the 1850 time slice value would be more standard practice. If the authors want to keep absolute values, my point was that it would be useful to evaluate the models against (say) a reanalysis product. Incidentally, given the biases in both the underlying climate and the chemistry, why report absolute ozone concentrations?

We agree with the reviewer that models are likely biased but the difficulty here to show anomalies is to use the adequate reference. In this paper, we focus on the period between 1990 and 2100 and if we use a reanalysis product of the contemporary period to calculate the anomaly, it will not affect the trend found from 2000 and 2100. In Figure 6, the box-whisker plot of 2000 has to be taken as a reference which gives mean values similar to the contemporary period. The trend has to be understood by taking the 2000 period as the reference. We added this point into Figure 6 caption.

2. For Figure 9 the authors are arguing that they should not apply a field significance test. I would strongly argue that it is best practice to do this. The data are presented as a map, and therefore just relying on local significance is insufficient. As mentioned before, I suggest consulting Wilks (2016, BAMS, doi: 10.1175/BAMS-D-15- 00267.1).

We considered the field significance in our indication of significant as suggested by the reviewer. We use a field significance test (Benjamini and Hochberg, 1995; Wilks, 2006) that satisfied the false discovery rate (FDR) criterion with $\alpha_{FDR}$ = 0.10. The FDR method was performed using p values

from local Student t-test that was computed for each grid points with 95% confidence level. We added this point in the new version of the paper (see P6, L5-L7) and change consequently Figure 9 and adapt the text accordingly.

GENERAL COMMENTS

1. For consideration by the authors: The authors now cite Schnell et al. (2015) in the context of hourly ozone analysis, but I would still think that this present manuscript would be more widely useful if the hourly ozone results were analyzed. I appreciate that this is a substantial undertaking, and the manuscript could stand as it is with the monthly mean analysis, but I think it would make the manuscript more relevant.

The purpose of our paper is more focused on climate than air quality. For this main reason, we think the time scale of 1 month is sufficient for our analysis. We prefer to leave the manuscript as it is with the monthly mean analysis.

2. I'm sorry if this sounds like nitpicking, but I would re-encourage the authors to revisit the use of paragraphs. There are still paragraphs going over 20-30 lines (e.g., start of Sections 2.2, 3.1, 3.2 etc etc). New point, new paragraph!

Thank you for this remark. We separate the text from sections 2.2-4.5 into different paragraphs following the recommendation of the reviewer and revisit the text of all these sections. The corrections are highlighted in blue in the revised version of the paper.

SPECIFIC COMMENTS (incl. minor corrections)

P1, L8 (and elsewhere): You say the "behavior of the annual cycle" is good, but I think this is (still) ambiguous. I would say something like "the shape correlates but the values are biased".

We replace the sentence "The ensemble mean of ACCMIP models simulates very well the behavior of the annual cycle of surface ozone."

by (P1, L7-L9)

"The shape of the annual cycle of surface ozone simulated by ACCMIP models is similar to the observations one but the model values are biased high."

P3, L30: UM-CAM and STOC-HadAM3 produce meteorology without interaction with the concentrations of radiatively active species calculated by the chemistry scheme (NOT "without interaction with climate")

We replace the sentence "which produce their own meteorology without any interaction with climate"

by (P3, L30-L31)

"which produce their own meteorological fields with no interaction with the concentrations of radiatively active species calculated by the chemistry scheme."

P4, L10: "models, however" -> "models; however" or "models. However" (Bad style to use "however" to start a new clause in the middle of the sentence)

Done (P4, L12)

P5, L29: I'm not sure about the "IQR for outliers". A sample outside of the central 50% is not generally considered an "outlier". There are several definitions of course, but perhaps you could use something like Tukey's Fences (see here for an introduction: https://en.wikipedia.org/wiki/Outlier)

In fact we used the method of Tukey-s Fences with a coefficient of 1.5. Outliers are defined as any values <25th percentile value–1.5 IQR (interquartile range) or values >75th percentile value + 1.5 IQR. This is clarified now in the revised version (see P6, L2-L3).

P7, L34: "Moreover" -> "However" or "But" (since introducing a counterpoint)

Done (P8, L18)

P13, L30: Be clear upfront that this is the chemical budget, excluding horizontal and vertical transport.

We replace the sentence "we focus on the evolution of the ozone budget along the 21st century"

by (P15, L15)

"we focus on the evolution of four ozone budget terms (excluding horizontal and vertical transport) along the 21st century"

P14, L7-8: I must have missed this the first time, but there are no strong reasons for basically defining an emergent constraint on the chemical budget changes ("models that are closest to the observations are the ones with increasing chemical terms". You might consider which models include climate-dependent biogenic emissions, which will have a strong impact on future emissions.

We removed this sentence that does not bring any added value to our paper.

Figure 5: Good that there is a color bar, but needs more description of how it was applied. Perhaps put the data in quintiles?

We added in the caption (Fig. 5) more description of how color bar was applied: "The colors associated with each metric value were determined as follows: the values of each metric have been rescaled between 0 and 1 corresponding to the model that is close to and far from the observations, respectively. The interval [0;1] has been subdivided into 6 equal intervals,

each representing a different color. The value of each metric is given by the color of the interval to which the rescaled value belongs."

Figure 6 (see also comment above): rainfall is normally reported as mm/day (convertible from what the models output)

As suggested by the reviewer, we changed the unit of the precipitation. It is now in mm/day in Fig. 6.

Responses to reviewer 2

We thank the reviewer for his/her valuable comments. We answer point to point to the comments in blue:

The authors generally address my comments. In particular, the authors improve the readability of the text; the english language of the text; and the conclusions. However, as the other referee notes, there is still room for improvement. I encourage the authors to address further these three points. The authors should also address the specific comments below.

Specific comments

P. 3 L. 13: I suggest: "…to contribute to the Intergovernmental…".
Done (P3, L13)

P. 8 L. 24: models -> model.
Done (P9, L10)

P. 9 L. 9: Style point: I suggest you do not start a sentence with an acronym.
Done  (P9, L30)

L. 20: Perhaps provide more details as to the effects mentioned and their impact on model results.

To clarify this point, we added the sentence (P10, L12) "More specifically, the increased humidity causes an ozone destruction which leads to a decrease in surface ozone."

 P. 12 L. 22: enhance -> enhances.
Done (P13, L34)

L. 24: I suggest you remove "indeed". Omit needless words.
Done

P. 13 L. 2: Could authors provide more details explaining this behaviour?

We provided details explaining this behavior in the conclusion of the same section (section 4.4) "For the RCP8.5, the future climate change associated with a net increase in CH4 concentration offsets the benefit of the emission reductions. In particular, for 2030 and 2100, the surface ozone concentration remains constant even if the NOx emissions are decreasing."

L. 21: Remind the reader in what ways is RCP8.5 atypical.

We replace the sentence

"The RCP8.5 is atypical and different from the other scenarios. The surface

ozone over the MB remains constant over the period 2000-2100 with a strong increase in temperature, specific humidity and CH4 concentration, unlike the global tropospheric ozone, which should increase by 18% in 2100 (Young et al., 2013)."

by (P15, L3-L5):

"For the RCP8.5 scenario, the surface ozone over the MB remains constant over the period 2000-2100 with a strong increase in temperature, specific humidity and CH4 concentration, unlike the global tropospheric ozone, which should increase by 18% in 2100 (Young et al., 2013)."

P. 15 L. 7: For style, I suggest that you use "first" as you use "second" later. Done (P17, L7)

L. 20: In what way is this change "non-significant"?

To clarify this point we change the sentence: "non-significant change" by "statistically non-significant changes" (P17, L20-L21).

 P. 33 Fig. 12: Please indicate in the caption the period over which you calculate the future relative change.

In the caption of Fig 12, we added the sentence: "The future relative change was calculated over the periods 2027-2040 and 2085-2110 (see Table. 3)."

---

## Author Response (AR4)

Responses to the editor

We thank the editor for his valuable comments. We answer point to point to the comments in blue:

Please address the following specific comments:

**P. 1**

L. 8: I suggest "...is similar to the annual cycle of the ozone observations, but..."
Done

L. 14: Introduce NOx, as you do on p.2.

We replace "the Nox-limited regime" by "the nitrogen oxide $(NO_x=NO+NO_2)$-limited regime"

L. 16: I suggest "...the benefits from the reduction..."
Done in the abstract and the entire paper.

**P. 2**

L. 2: Assessments of what? Chemical and dynamical processes in the atmosphere?
We removed the term « and assessments » to clarify this point.

L. 5: "...of the residual..."
Done

L. 21: What exactly do you mean by "hot-spot"?

We change the sentence containing « hotspot » by : « This region is sensitive to climate change (Giorgi, 2006) that is due to its particular location and diversity of ecosystems."

L. 26: changes. Changes owing to what?
We replace the sentence "the future change of surface ozone in Europe" by "the future changes of surface ozone due to climate change and ozone precursors evolution"

L. 29: I suggest the form "..., e.g., ..."
Done

L. 33: The assessment -> An assessment
Done

L. 34: "At the regional scale…"
Done

**P. 3**

L. 4: Omit "the"

Done

L. 7: "…emissions, and meteorological…"
Done

L. 12 and elsewhere: I suggest you do not start a sentence with an acronym
replace the "ACCMIP" by "This intercomparison project (ACCMIP)"

L. 13-14: Perhaps it would be better to write. "…and analyses the driving forces…"
Done (see p3, L15)

L. 18: changes
Done (see p3, L19)

L. 19: these changes

Done (see p3, L20)

L. 26+: Introduce acronyms for the models when first used

Most of the model's names is a mix of name from their lab and different model, which could make the text unclear. However, we introduce the acronyms for the models as much as necessary in the revised version of the paper. In addition, we added the following reference which give many information on the different models used in our paper:  Lamarque et al. 2013 : "https://www.geosci-model-dev.net/6/179/2013/gmd-6-179-2013-supplement.pdf"

**P. 4**

L. 17: "…according to the radiative…"
Done (see p4, L20)

L. 24: I suggest "…), and one very…"
Done (see p4, L27)

L. 27: Secondly -> Second. To match the use of "first" earlier
Done (see P4, L30)

**P. 5**

L. 14: I suggest you use "one" and "four" instead of "1" and "4"
Done (see P5, L17)

**P. 6**

L. 6: "…from a local…"
Done (see P6, L10)

L. 25: Is the bias positive? If so, indicate
Done (see P6, L30)

**P. 7**

L. 13: For multiple citations order the citations by alphabetical or chronological order but do not mix. In particular, for citations in the same year, order alphabetically
Done (see P7, L18-19)

L. 23: Introduce the acronym for "sd" – you do so in P. 13
The acronym has already been defined just earlier (P7, L25)

**P. 8**

L. 27: Do you really mean global scales? Earlier, you suggest you focus on the Mediterranean Basin. Please clarify
In our study, we compare the performances of the different models for the regional MB scale vs the global scale.

To clarify this we rewrote this sentence by (see P8, L30-31) "In our study, we use the ACCMIP simulations of surface ozone over a specific region,

namely over the MB, but we compare the performances of the models at the regional MB and global scales »

L. 5: "…that the CMAM…"
Done (see P9, L9)

L. 8: Perhaps better to write "…bias is positive at the regional and global scales for all models except…" Is this what you mean?
Done (see P9, L12)

L. 10: "…in the recent past…"
Done (see P9, L14)

L. 15: "…and future…"
Done (see P9, L19)

L. 26: "…The CH4…" I suggest you do not start sentences with an abbreviation or acronym
Done (see P10, L31)

L. 2: "Tukey's fences rule…" Omit "the"
Done (see P11, L7)

L. 7: bias -> biased
Done (see P12, L12)

L. 16: I suggest you mention first the earlier year, e.g., "2000 (REF) and 2030"
Done (see P12, L21)

L. 4: "…due to evaporation…"
Done (see P13, L8)

L. 23: showed -> show
Done (see P13, L26)

**P. 14**

L. 18: has -> have
Done (see P14, L23)

L. 20: "…is similar in magnitude for the two scenarios…"
Done (see P14, L25)

L. 26: Quantify the "intense" mentioned. "Latter" what?

We change the sentence "We note that, for the RCP8.5, the change in surface ozone over the MB is less intense than the global tropospheric ozone change."

by (see P14, L30-32)

We note that, for the RCP8.5, the relative changes in summer surface ozone in 2030 (2100) over the MB is less intense with values of -1.3% (-0.8%) than for the global tropospheric ozone change with values of 7% (18 %). This global tropospheric ozone change has already been highlighted by Young et al. (2013).

**P. 15**

L. 9: "…combined with a…"

Done (see P15, L16)

L. 28: "For the 2030…"

Done (see P16, L1)

**P.16**
L. 4: Could you quantify this proportional behaviour?

We replace "Dry deposition of ozone decreases for all scenarios from 2030 to 2100, in a proportional way to that of the surface ozone."

by (see P16, L10-12)

"For all scenarios from 2030 to 2100, dry deposition of ozone decreases like

surface ozone concentration with a more pronounced decrease for RCP2.6 and RCP6.0 than for RCP4.5"

L. 7: You mean that the net effect is one of randomness. Clarify

To clarify we change the sentence by (see P16, L13-15) "For the RCP8.5, the surface ozone budget terms of each model evolve differently which explains the non-significant changes in surface ozone and its stagnation over the MB."

**P. 17**

L. 22: You suggest these other results are significant. If so, at which level?

We changed the sentence "which show an ozone decrease"
by (see P17, L22)
"Which show a statistically significant ozone decrease (using the Student t-test with 95% confidence level)"

L. 23-24: Quantify the "intensively" and "less strongly" qualifications

We replace the sentence "… decreases intensively from 2030 to 2100 for RCP2.6, RCP4.5 and RCP6.0 and less strongly for the RCP8.5"
by (see P17, L24)
"… decreases intensively (15%-25%) from 2000 to 2100 for RCP2.6, RCP4.5 and RCP6.0 and less strongly (10%) for the RCP8.5"

L. 26+ paragraph: Quantify the decreases mentioned
Done (see P17, L24-27)

L. 31: I suggest you avoid the use of "etc"
Done

**P. 18**

L. 3-4: Indicate how this builds on the work done in this paper.
We removed this sentence that does not bring any added value to our paper.

L. 8: improve -> improved
Done (see P18, L9)

**P. 26**

Fig. 2: thin line -> thin lines.
Done

Mention in the caption the legends in the two figures
We added in the caption the following sentence "The different models and the observations are represented by a color as shown in the legends of each figure."

**P. 30**

Fig, 6: Please rephrase the penultimate sentence; it is not clear to me

We replaced the following sentence "The median is indicated by the thick horizontal black line, the multi model mean by a filled diamond, the (25-75%) range by the colored box and minimum/maximum excluding outliers by whisker. "

by

« The median is indicated by the horizontal black solid line and the multi model mean by a filled black diamond. The range (25-75%) is represented by the length of each colored box and the minimum/maximum (excluding outliers) by the whisker. »

**P. 33**

Fig. 9: When talking about the anomalies, I suggest you mention first the earlier year, e.g., "2030-2100"
Done

**P. 35**

Fig. 11: "…the multi-model mean…"; "…by the whisker…". Same for Fig. 12.
Done